# Diatom Flora from Time-Series Sediment Trap in the Kuroshio Extension Region of the Northwestern Pacific



**Joon Sang Park [1,\*], Hyung Jeek Kim [2], Kyun-Woo Lee [3], Hyun Ju Ha [1] and Yun Jae Kim [3]**

[1] Library of Marine Samples, Korea Institute of Ocean Science & Technology, Geoje 53201, Republic of Korea; calltolove@kiost.ac.kr

[2] Tropical & Subtropical Research Center, Korea Institute of Ocean Science & Technology, Jeju 63349, Republic of Korea; juac29@kiost.ac.kr

[3] Marine Biotechnology & Bioresource Research Department, Korea Institute of Ocean Science & Technology, Busan 49111, Republic of Korea; kyunu@kiost.ac.kr (K.-W.L.); bio1212@kiost.ac.kr (Y.J.K.)

\* Correspondence: jspark1101@kiost.ac.kr

**Abstract:** Precise identification of diatom species is fundamental to correctly interpreting their roles in the marine ecosystems; the documentation of species records with illustrations is therefore essential to guarantee ecological works and the continuous use of compositional data in future works. We document the diatom flora in the Kuroshio Extension (KE) area of the northwestern Pacific. Samples were collected by sediment trap deployment from November 2017 to August 2018 and identified using light microscopy and scanning electron microscopy. Eighty-two taxa belonging to 17 families and 38 genera were documented with representative references, morphological dimensions, brief diagnosis, distribution, and short taxonomic comments. All of the taxa were divided into three distribution patterns (cold, warm, and eurythermal taxa) based on the previous distribution records; each group may be transported by the Oyashio and Kuroshio Currents, respectively. The mixed occurrence of cold- and warm-water species indicates that the KE area is a crossroads for them. A preliminary checklist was compiled from previous studies, incorporating our records, and 206 diatom taxa occurred in the northeastern path of the Kuroshio Current. The diatom flora in the KE area will be used to understand the hydrology of the Kuroshio Current in future work.

**Keywords:** bioindicator; hydrological proxy; Kuroshio Current; Oyashio Current; species diversity

## 1. Introduction

Diatoms play a crucial role in the ocean as the main primary producers and transporters of particulate organic matter into the ocean's interior. Although diatoms are not abundant in the oligotrophic oceans of the subtropical gyre [1–3], they might be more important for new production and the export of particulate carbon than previously thought in oligotrophic oceans [4,5]. In particular, episodic nutrient input into oligotrophic regions leads to rapid diatom growth and a coupled export flux [4,6]. There are many time-series sediment trap studies, most of which have focused on geochemical and gross uptake and export studies for carbon or silica cycles (e.g., [7]). Before understanding the dynamics of phytoplankton communities under environmental stress, documentation of the flora is essential for properly understanding changes in a dynamic marine environment. Morphological works should steadily provide information despite being time-consuming and requiring effort. Despite the large number of sediment trap studies associated with phytoplankton ecology, the use of data derived from sediment traps needs more support (e.g., [8]). In addition, studies on the flora of diatoms from sediment traps have received little attention, despite their importance for the insurance of species identification and evaluation of diatom assemblage structures between regions [9].

The Kuroshio Extension (KE) is a boundary area between the northward warm Kuroshio Current and the southward cold Oyashio Current, and the marine environ-

ment is relatively unstable due to complex hydrodynamic processes such as large-scale recirculation and mesoscale eddies (e.g., [10]). The phytoplankton composition in this area reflected its instability, showing changes in community composition in space and time (e.g., [11]). Several studies regard KE-adjacent regions as a "hotspot" of primary production with significant carbon sinks [12,13]. A time-series sediment trap was deployed in the KE region between November 2017 and August 2018 to understand the flux and sinking particles from the KE region [14], and the components (in particular, the diatoms) of the sinking flux were analyzed. In order to guarantee the diatom flux data quality, we provide the diatom flora that comprise the trap samples with illustrations and a brief description of the species.

## 2. Materials and Methods

### 2.1. Processing of the Sediment Trap

A time-series sediment trap (PARFLUX Mark 78H-21, McLane, Falmouth, MA, USA) was deployed at station KE08 (Figure 1, 33°41.8′ N, 156°39.7′ E) at a depth of 800 m from November 2017 to August 2018. A recording current meter (SeaGuard RCM DW, Aanderaa, Bergen, Norway) for the measurement of current speed and direction was deployed at an 835 m water depth (35 m below the sediment trap). A total of 20 bottles were collected over 10 months, but the 16th bottle was lost. Sinking particles were collected at 10-day intervals from February to June 2018 and monthly during the remaining time. Detailed sample intervals and information were presented by Kim et al. [14]. Sample bottles were filled with filtered seawater collected from the trap deployment depth at the same site. Sodium-borate-buffered, 5% formalin solution was added as a preservative. Upon recovery, samples were stored in a refrigerator at 2–4 °C.

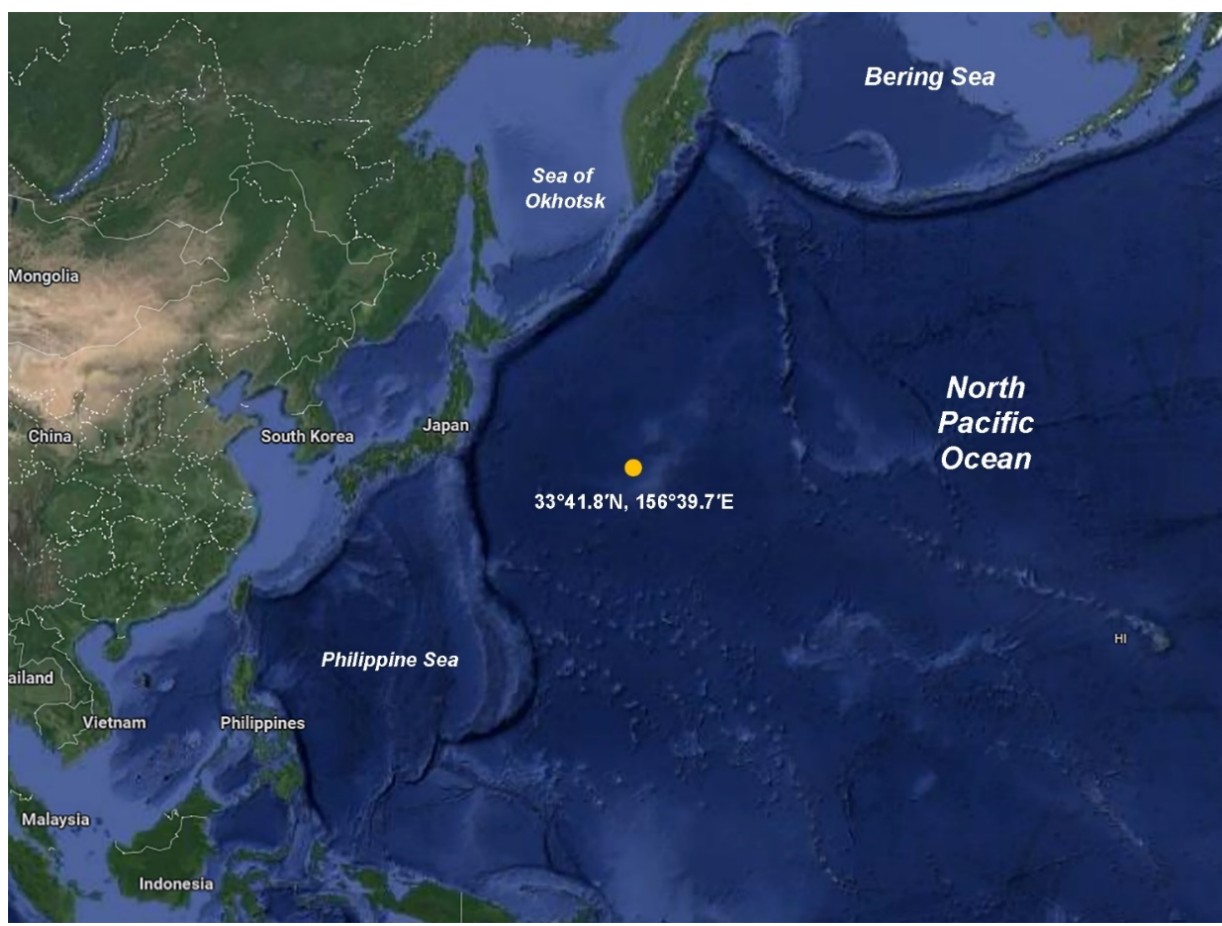

**Figure 1.** Sediment trap deployment site in the Kuroshio Extension area (yellow circle).

*2.2. Diatom Analysis*

The original samples were rinsed with preservatives and seawater, and the organic materials in the sample pellets were removed following Hasle and Fryxell [15]. In brief, equal amounts of saturated potassium permanganate and 33% hydrochloric acid were added and boiled at 80 °C until the color was cleared. The cleaned samples were rinsed until free of acid. Preparation of slides for qualitative and quantitative analyses involved the settling of the acid-cleaned material onto cover slips by a random settling method [16]. The cover slips were affixed with a Pleurax (Mountmedia, Wako, Japan). The permanent slides were used for qualitative analysis of diatom under a ×1000 objective using a light microscope (BX51; Olympus, Tokyo, Japan) with a digital camera (EOS 50D; Cannon, Seoul, Korea). For scanning electron microscopy (SEM) examinations, some cleaned materials were filtered through a 0.2 polycarbonate membrane and dried in the air. The filtrates were attached to aluminum stubs and coated by 10 nm gold–palladium. The prepared stubs were observed in FE-SEM (JSM7600F, Jeol, Japan). For the identification of the species, we referred to various studies, including ones on the quantitative and qualitative morphological characteristics and distributions of the species. The references are provided in each species section with description pages and figure numbers.

## 3. Results

A total of 82 diatom taxa were observed in trap samples (Table 1, Figures 2–12). The references for species identification, brief diagnostics, micrographs, and dimensions for all diatoms are presented. Higher classification ranking into subclasses follows Mann in Adl et al. [17], and the order levels followed Algaebase [18].

**Table 1.** List of diatoms and the currents associated with their occurrences.

| Taxa | Kuroshio | Oyashio | Uncertain |
|---|:---:|:---:|:---:|
| **Subphylum Coscinodiscophytina** | | | |
| **Class Coscinodiscaceae** | | | |
| **Subclass Coscinodiscophycidae** | | | |
| **Order Asterolamprales** | | | |
| **Family Asterolampraceae** | | | |
| **Genus *Asteromphalus*** | | | |
| *Asteromphalus flabellatus* | + | | |
| *Asteromphalus heptactis* | | + | |
| **Genus *Spatangidium*** | | | |
| *Spatangidium arachne* | + | | |
| **Order Coscinodiscales** | | | |
| **Family Coscinodiscaceae** | | | |
| **Genus *Coscinodiscopsis*** | | | |
| *Coscinodiscopsis jonesiana* | + | | |
| **Genus *Coscinodiscus*** | | | |
| *Coscinodiscus argus* | | | + |
| *Coscinodiscus centralis* | + | | |
| *Coscinodiscus gigas* | + | | |
| *Coscinodiscus* cf. *marginatus* | | | + |
| *Coscinodiscus radiatus* | | | + |
| **Family Heliopeltaceae** | | | |
| **Genus *Actinoptychus*** | | | |
| *Actinoptychus senarius* | + | | |

**Table 1.** *Cont.*

| Taxa | Kuroshio | Oyashio | Uncertain |
|------|:--------:|:-------:|:---------:|
| **Family Hemidiscaceae** | | | |
| **Genus *Actinocyclus*** | | | |
| *Actinocyclus curvatulus* | | + | |
| *Actinocyclus iraidae* | | + | |
| *Actinocyclus ochotensis* | | + | |
| *Actinocyclus octonarius* | | | + |
| **Genus *Azpeitia*** | | | |
| *Azpeitia neocrenulata* | + | | |
| *Azpeitia nodulifera* | + | | |
| **Genus *Roperia*** | | | |
| *Roperia tesselata* | + | | |
| **Order Stellarimales** | | | |
| **Family Stellarimaceae** | | | |
| **Genus *Stellarima*** | | | |
| *Stellarima stellaris* | + | | |
| **Subphylum Rhizosoleniophytina** | | | |
| **Class Rhizosoleniophyceae** | | | |
| **Order Rhizosoleniales** | | | |
| **Family Rhizosoleniaceae** | | | |
| **Genus *Pseudosolenia*** | | | |
| *Pseudosolenia calcar-avis* | + | | |
| **Genus *Rhizosolenia*** | | | |
| *Rhizosolenia bergonii* | + | | |
| *Rhizosolenia hebetata* f. *semispina* | | + | |
| *Rhizosolenia imbricata* | + | | |
| *Rhizosolenia styliformis* | | | + |
| **Genus *Sundstroemia*** | | | |
| *Sundstroemia pungens* | + | | |
| **Subphylum Probosciophytina** | | | |
| **Class Probosciophyceae** | | | |
| **Order Probosciales** | | | |
| **Family Probosciaceae** | | | |
| **Genus *Proboscia*** | | | |
| *Proboscia indica* | + | | |
| **Subphylum Bacillariophytina** | | | |
| **Class Mediophyceae** | | | |
| **Subclass Thalassiosirophycidae** | | | |
| **Order Thalassiosirales** | | | |
| **Family Lauderiaceae** | | | |
| **Genus *Lauderia*** | | | |
| *Lauderia annulata* | + | | |
| **Family Thalassiosiraceae** | | | |
| **Genus *Detonula*** | | | |
| *Detonula confervacea* | | + | |
| **Genus *Minidiscus*** | | | |
| *Minidiscus trioculatus* | | | + |
| **Genus Planktoniella** | | | |
| *Planktoniella blanda* | + | | |
| *Planktoniella sol* | + | | |

**Table 1.** *Cont.*

| Taxa | Kuroshio | Oyashio | Uncertain |
|---|---|---|---|
| **Genus *Shionodiscus*** | | | |
| *Shionodiscus oestrupii* var. *oestrupii* | + | | |
| *Shionodiscus oestrupii* var. *venrickae* | + | | |
| *Shionodiscus poro-irregulatus* | | | + |
| *Shionodiscus trifultus* | | + | |
| *Shionodiscus variantus* | | + | |
| **Genus *Takanoa*** | | | |
| *Takanoa bingensis* | | | + |
| **Genus *Thalassiosira*** | | | |
| *Thalassiosira anguste-lineata* | | | + |
| *Thalassiosira diporocyclus* | + | | |
| *Thalassiosira ferelineata* | + | | |
| *Thalassiosira lineata* | + | | |
| *Thalassiosira mendiolana* | + | | |
| *Thalassiosira punctifera* | + | | |
| *Thalassiosira sacketii* | + | | |
| *Thalassiosira subtilis* | + | | |
| *Thalassiosira symmetrica* | + | | |
| *Thalassiosira tenera* | + | | |
| **Subclass Lithodesmiophycidae** | | | |
| **Order Lithodesmiales** | | | |
| **Family Lithodesmiaceae** | | | |
| **Genus *Lithodesmium*** | | | |
| *Lithodesmium variabile* | | | + |
| **Subclass Chaetocerotophycidae** | | | |
| **Order Chaetocerotales** | | | |
| **Family Chaetocerotaceae** | | | |
| **Genus *Bacteriastrum*** | | | |
| *Bacteriastrum elongatum* | + | | |
| *Bacteriastrum furcatum* | + | | |
| **Genus *Chaetoceros*** | | | |
| *Chaetoceros affinis* | + | | |
| *Chaetoceros atlanticus* | | + | |
| *Chaetoceros didymus* | + | | |
| *Chaetoceros eibenii* | + | | |
| *Chaetoceros messanensis* | + | | |
| *Chaetoceros peruvianus* | + | | |
| *Chaetoceros radicans* | | | + |
| **Order Hemiaulales** | | | |
| **Family Hemiaulaceae** | | | |
| **Genus *Cerataulina*** | | | |
| *Cerataulina pelagica* | + | | |
| **Genus *Eucampia*** | | | |
| *Eucampia cornuta* | + | | |
| **Genus *Hemiaulus*** | | | |
| *Hemiaulus sinensis* | + | | |

**Table 1.** *Cont.*

| Taxa | Kuroshio | Oyashio | Uncertain |
|---|---|---|---|
| **Class Bacillariophyceae** | | | |
| **Subclass Fragilariophycidae** | | | |
| **Order Thalassionematales** | | | |
| **Family Thalassionemataceae** | | | |
| **Genus *Lioloma*** | | | |
| *Lioloma pacificum* | + | | |
| **Genus *Thalassionema*** | | | |
| *Thalassionema bacillare* | + | | |
| *Thalassionema frauenfeldi* | + | | |
| *Thalassionema kuroshioensis* | + | | |
| *Thalassionema nitzschioides* var. *nitzschioides* | + | | |
| *Thalassionema nitzschioides* var. *parva* | + | | |
| **Subclass Urneidophycidae** | | | |
| **Order Rhaphoneidales** | | | |
| **Family Rhaphoneidaceae** | | | |
| **Genus *Neodelphineis*** | | | |
| *Neodelphineis indica* | + | | |
| **Subclass Bacillariophycidae** | | | |
| **Order Naviculales** | | | |
| **Family Naviculaceae** | | | |
| **Genus *Navicula*** | | | |
| *Navicula* cf. *transistantioides* | | | + |
| **Family Pleurosigmataceae** | | | |
| **Genus *Pleurosigma*** | | | |
| *Pleurosigma directum* | | | + |
| *Pleurosigma diversestriatum* | + | | |
| **Order Bacillariales** | | | |
| **Family Bacillariaceae** | | | |
| **Genus *Alveus*** | | | |
| *Alveus marinus* | + | | |
| **Genus *Fragilariopsis*** | | | |
| *Fragilariopsis* aff. *oceanica* | | + | |
| *Fragilariopsis doliolus* | + | | |
| **Genus *Neodenticula*** | | | |
| *Neodenticula seminae* | | + | |
| **Genus *Nitzschia*** | | | |
| *Nitzschia bicapitata* | + | | |
| *Nitzschia bifurcata* | + | | |
| *Nitzschia interruptestriata* | + | | |
| *Nitzschia kolaczeckii* | + | | |
| *Nitzschia sicula* var. *sicula* | + | | |
| *Nitzschia sicula* var. *bicuneata* | + | | |
| **Genus *Psammodictyon*** | | | |
| *Psammodictyon* sp. | | | + |
| **Genus *Pseudo-nitzschia*** | | | |
| *Pseudo-nitzschia turgiduloides* | | + | |
| **Genus *Tryblionella*** | | | |
| *Tryblionella coarctata* | + | | |
| | 56 | 12 | 14 |

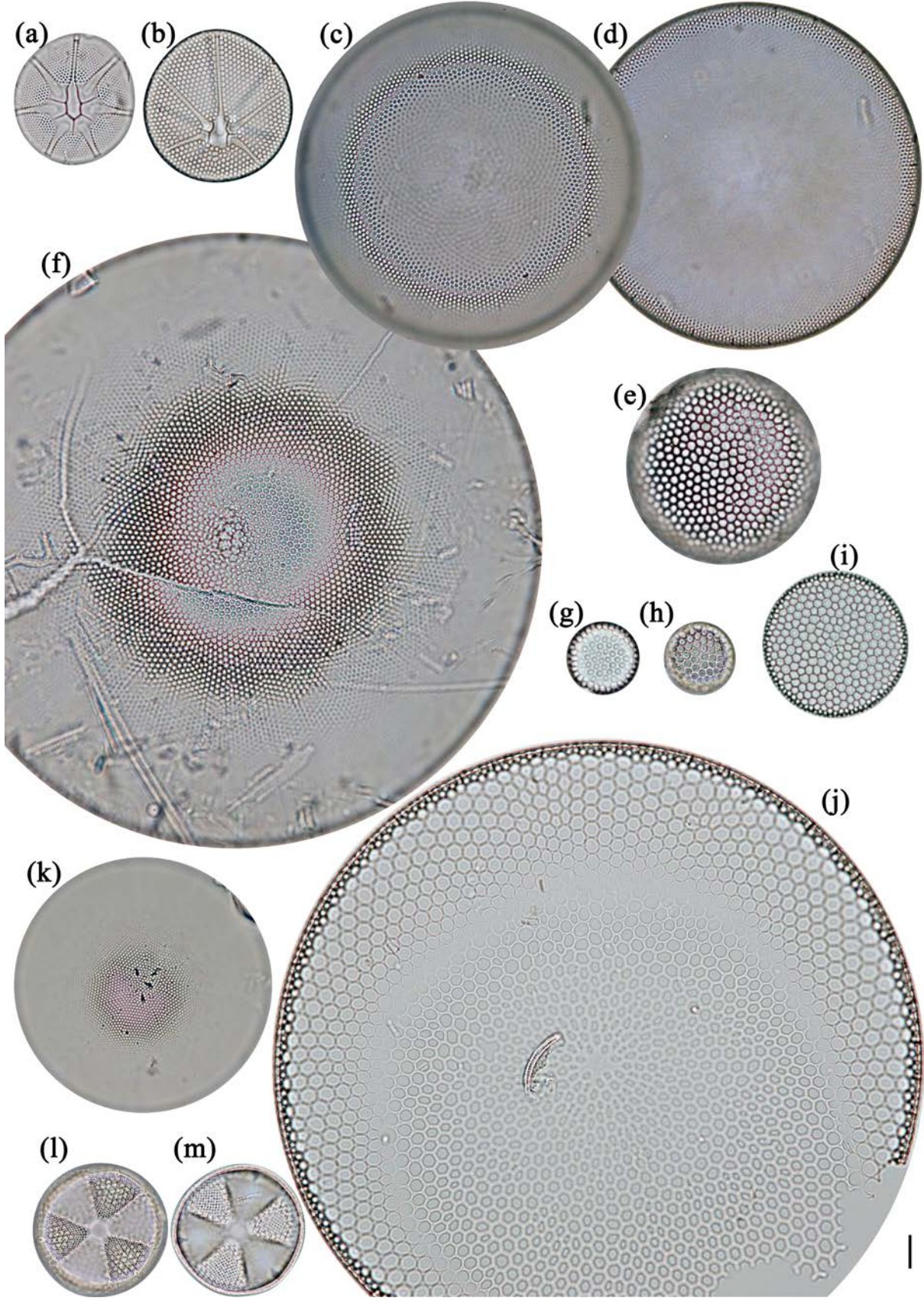

**Figure 2.** (**a**) *Asteromphalus heptactis*; (**b**) *Spatangidium arachne*; (**c**,**d**) *Coscinodiscopsis jonesiana*; (**e**) *Coscinodiscus argus*; (**f**) *Coscinodiscus centralis*; (**g**,**h**) *Coscinodiscus* cf. *marginatus*; (**i**) *Coscinodiscus radiatus*; (**j**) *Coscinodiscus gigas*; (**k**) *Stellarima stellaris*; (**l**,**m**) *Actinoptychus senarius*. Scale bar = 10 μm.

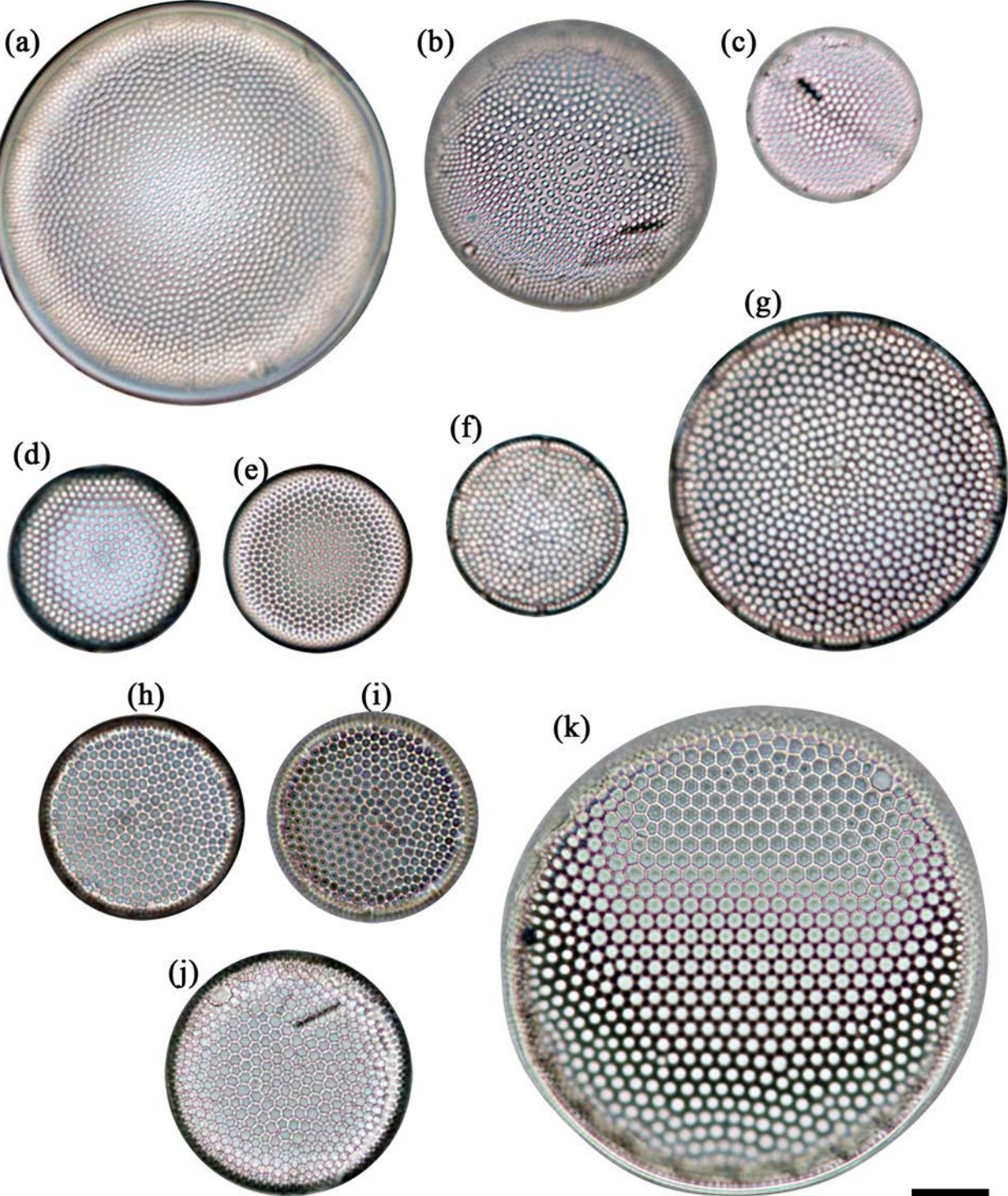

**Figure 3.** (**a**) *Actinocyclus octonarius*; (**b**) *Actinocyclus ochotensis*; (**c**) *Actinocyclus curvatulus*; (**d**,**e**) *Actinocyclus iraidae*; (**f**,**g**) *Azpeitia neocrenulata*; (**h**,**i**) *Azpeitia nodulifera*; (**j**,**k**) *Roperia tesselata*. Scale bar = 10 μm.

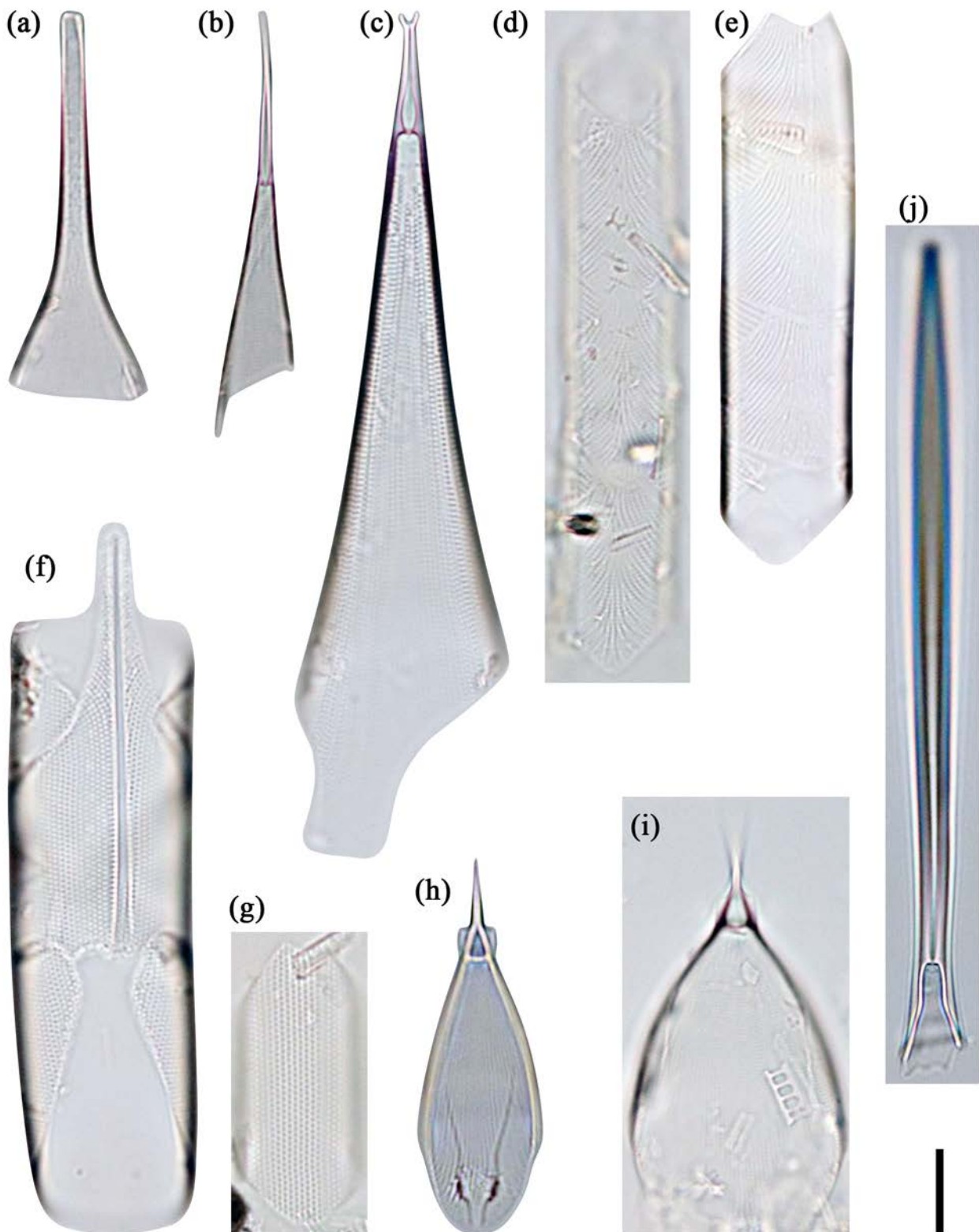

**Figure 4.** (**a**) *Proboscia indica*; (**b**) *Pseudosolenia calcar-avis*; (**c–e**) *Rhizosolenia bergonii*; (**f,g**) *Rhizosolenia hebetata* f. *semispina*; (**h**) *Rhizosolenia imbricata*; (**i**) *Rhizosolenia styliformis*; (**j**) *Sundstroemia pungens*. Scale bar = 10 μm.

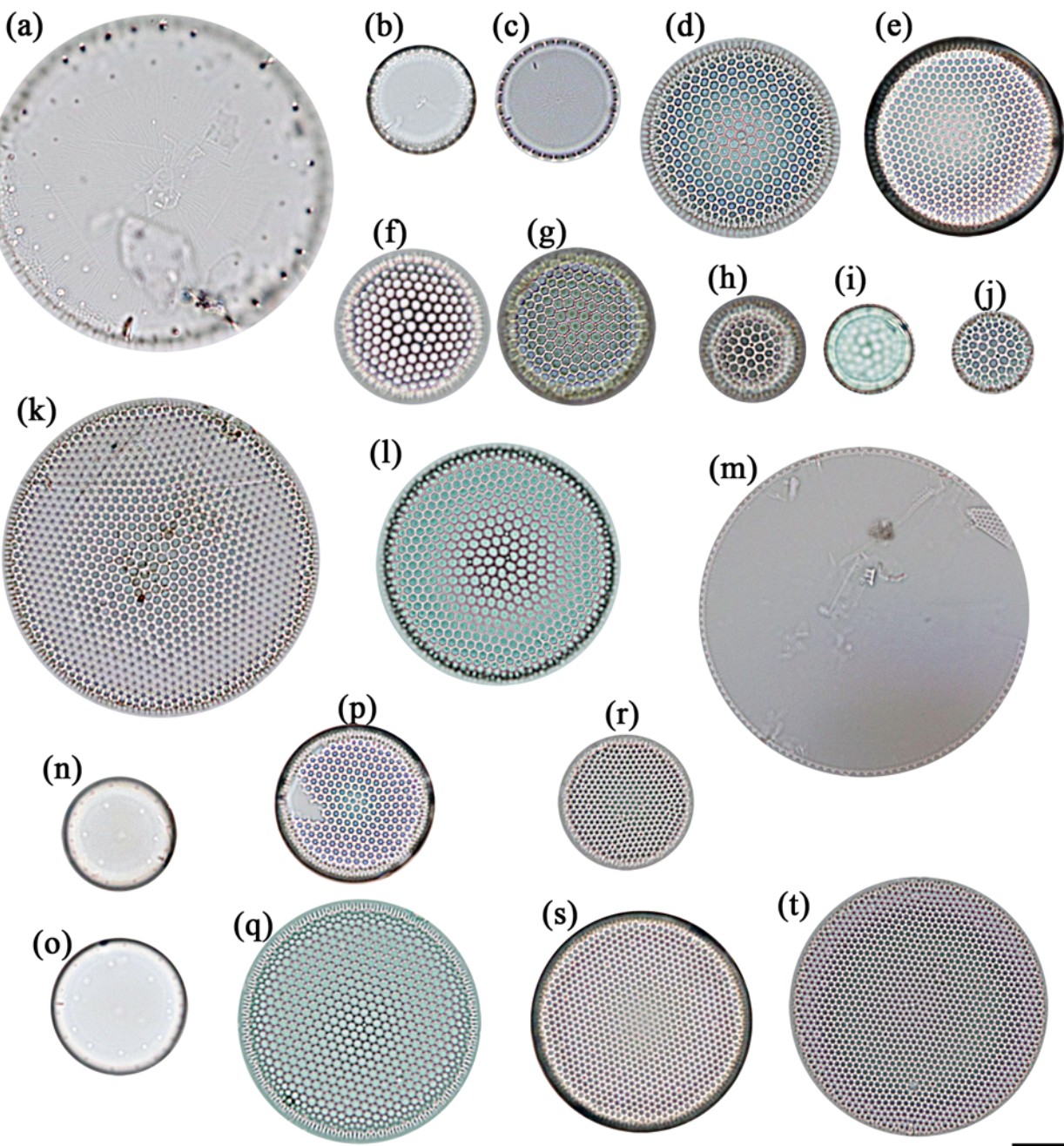

**Figure 5.** (**a**) *Lauderia annulata*; (**b**,**c**) *Detonula confervacea*; (**d**) *Planktoniella blanda*; (**e**) *Planktoniella sol*; (**f**–**j**) *Shionodiscus oestrupii* var. *oestrupii*; (**k**) *Shionodiscus* cf. *oestrupii* var. *venrickae*; (**l**) *Shionodiscus* aff. *poroirregulatus*; (**m**) *Takanoa bingensis*; (**n**,**o**) *Thalassiosira dioporocyclus*; (**p**,**q**) *Thalassiosira ferelineata*; (**r**–**t**) *Thalassiosira lineata*. Scale bar = 10 μm.

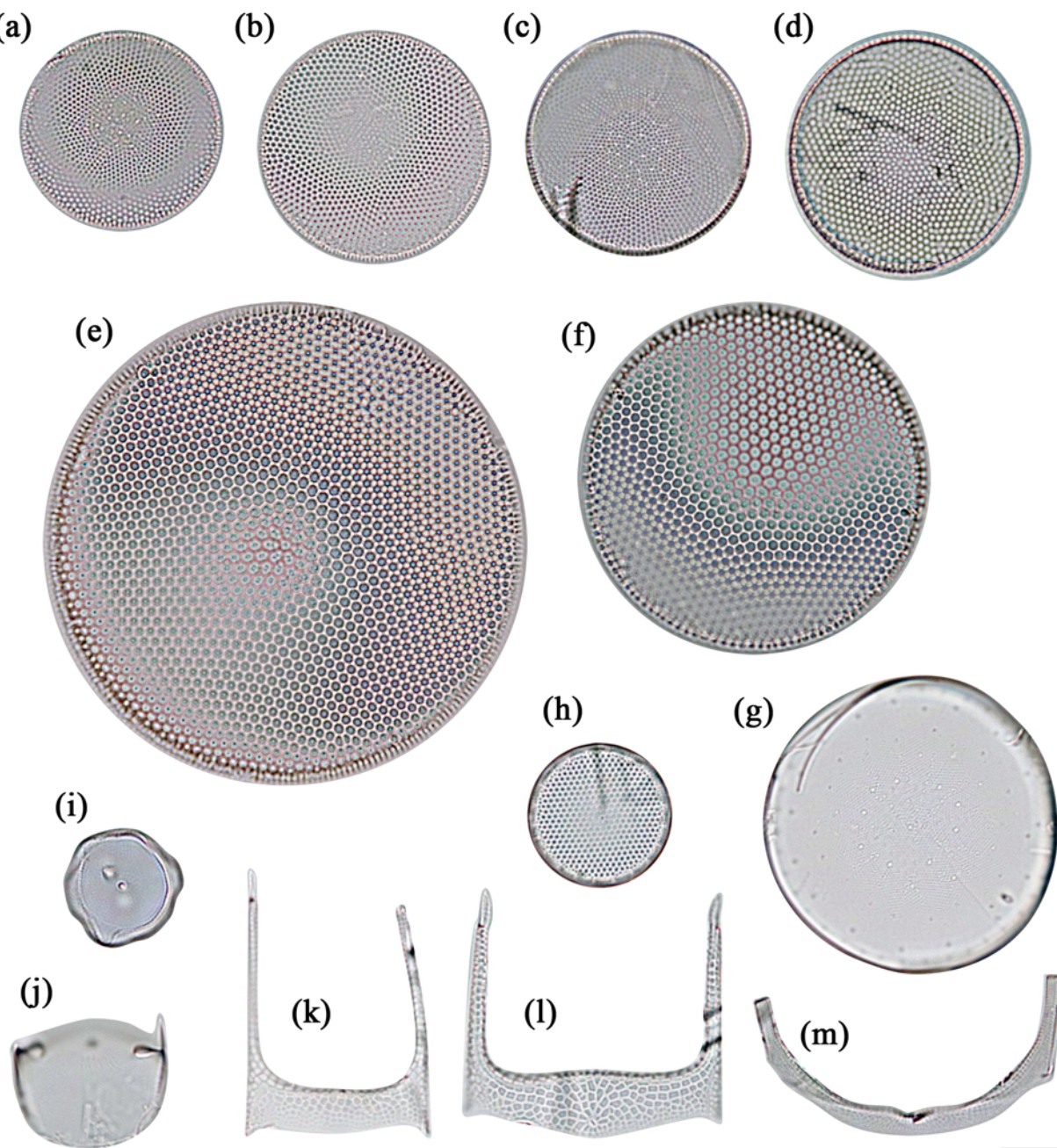

**Figure 6.** (**a**–**d**) *Thalassiosira sacketii*; (**e**,**f**) *Thalassiosira symmetrica*; (**g**) *Thalassiosira subtilis*; (**h**) *Thalassiosira tenera*; (**i**) *Lithodesmium variabile*; (**j**) *Cerataulina pelagica*; (**k**,**l**) *Hemiaulus sinensis*; (**m**) *Eucampia ornuta*. Scale bar = 10 μm.

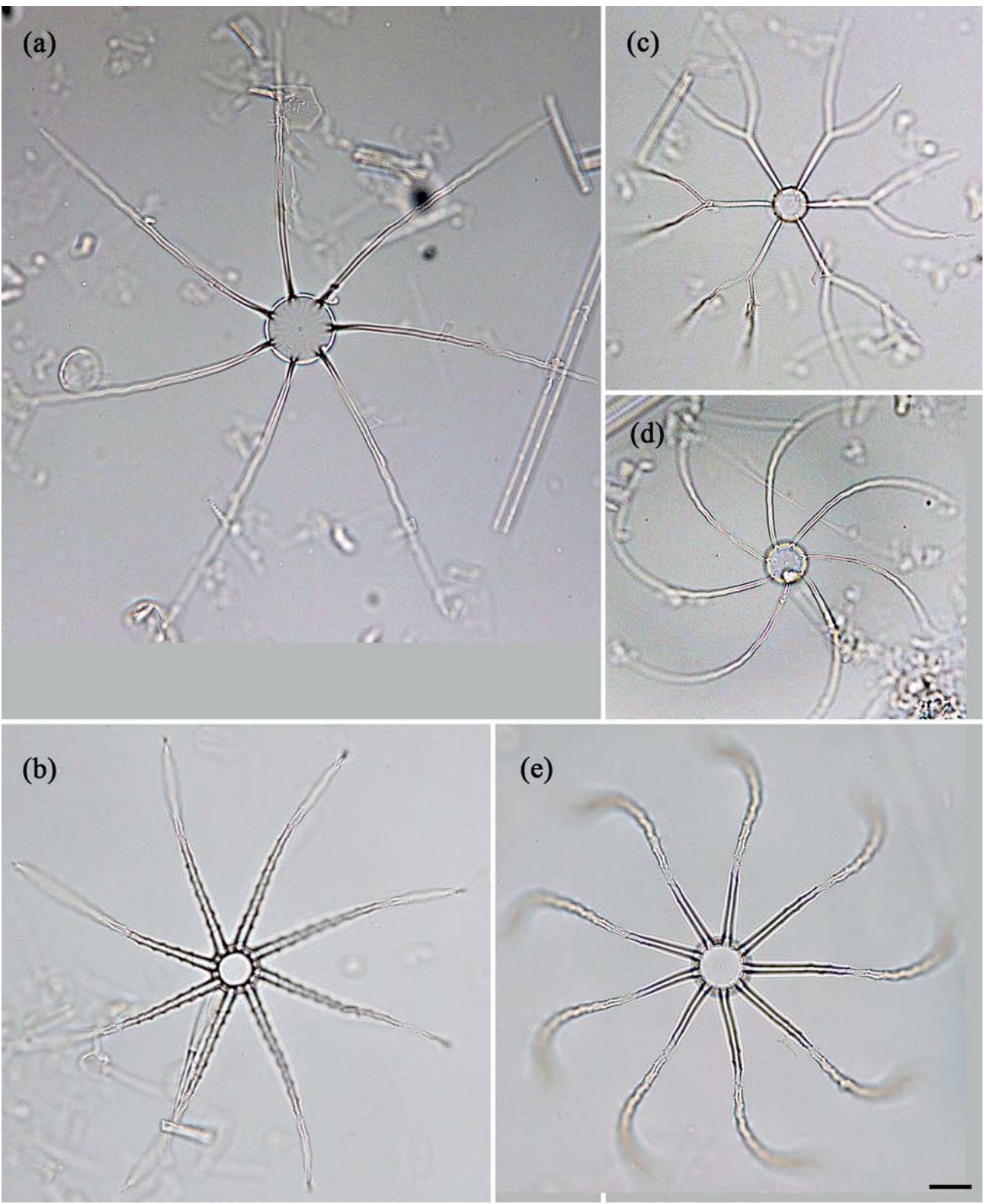

**Figure 7.** (**a**,**b**) *Bacteriastrum elongatum*; (**c**–**e**) *Bacteriastrum furcatum*. Scale bar = 10 μm.

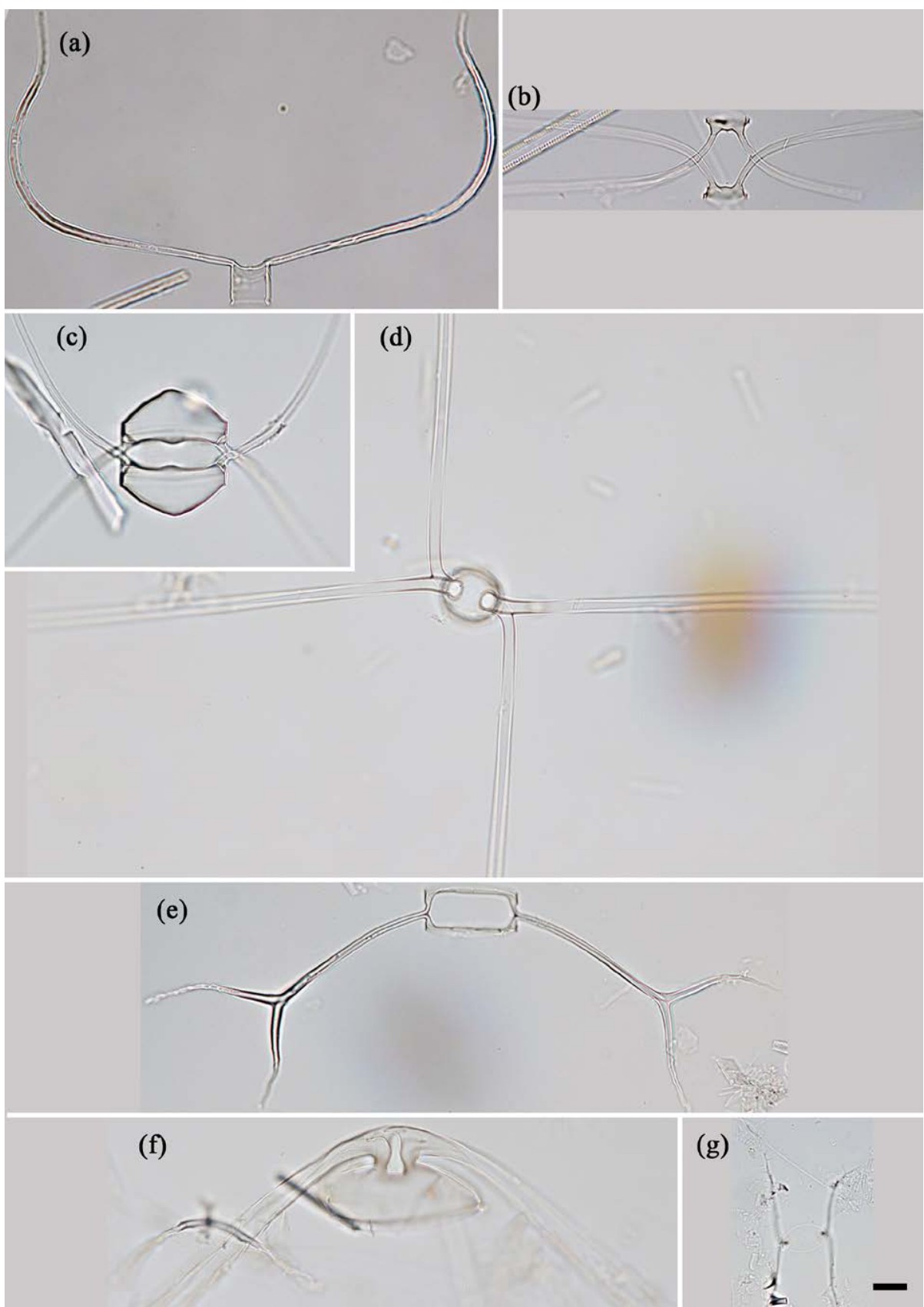

**Figure 8.** (**a**) *Chaetoceros affinis*; (**b**) *Chaetoceros atlanticus*; (**c**) *Chaetoceros didymus*; (**d**) *Chaetoceros eibenii*; (**e**) *Chaetoceros messanensis*; (**f**) *Chaetoceros peruvianus*; (**g**) *Chaetoceros radicans*. Scale bar = 10 μm.

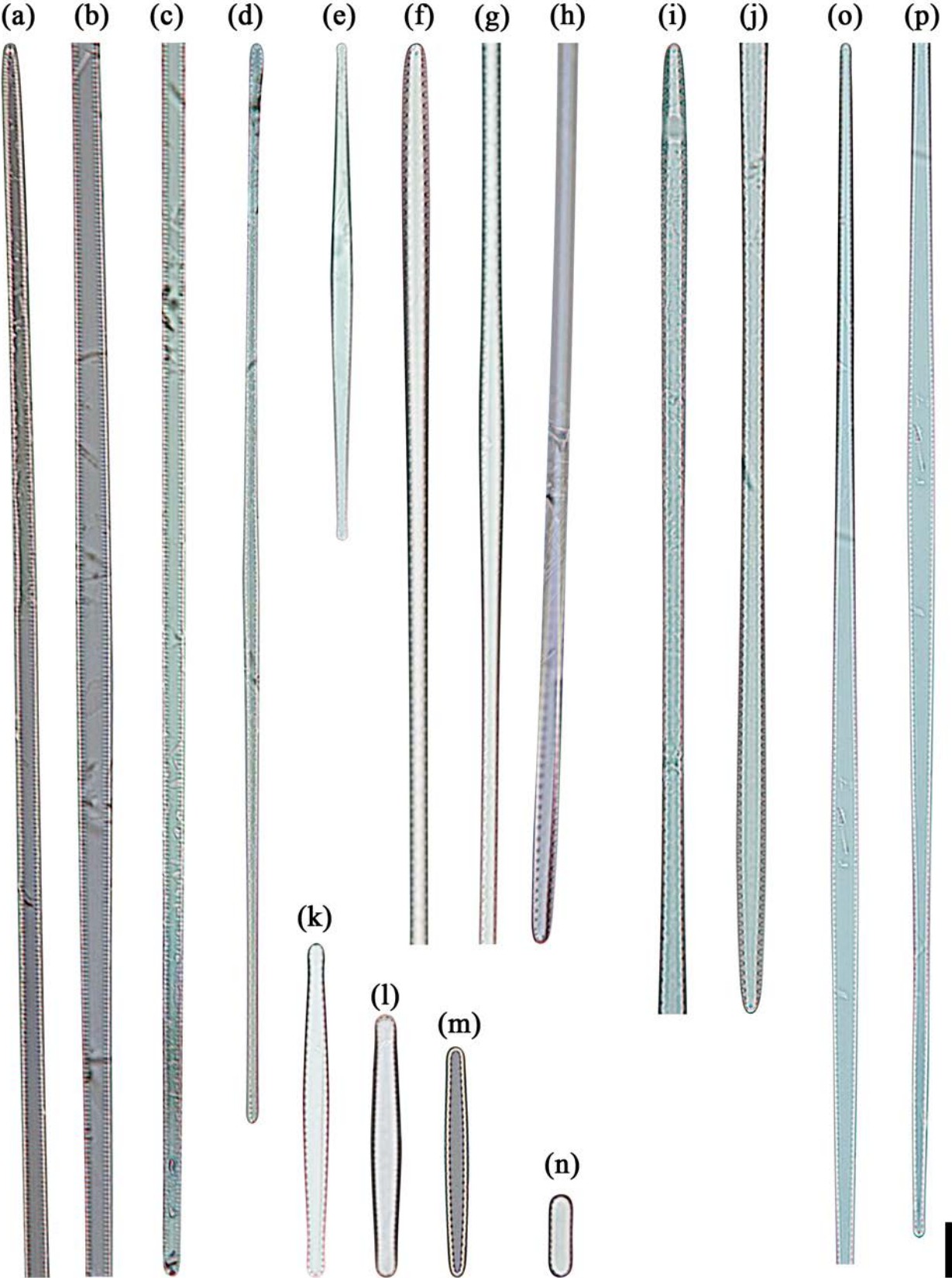

**Figure 9.** (**a**–**c**) *Lioloma pacificum*; (**d**,**e**) *Thalassionema bacillare*; (**f**–**j**) *Thalassionema frauenfeldii*; (**k**–**m**) *Thalassionema nitzschioides* var. *nitzschoides*; (**n**) *Thalassionema nitzschioides* var. *parva*; (**o**,**p**) *Thalassionema kuroshioensis*. Scale bar = 10 μm.

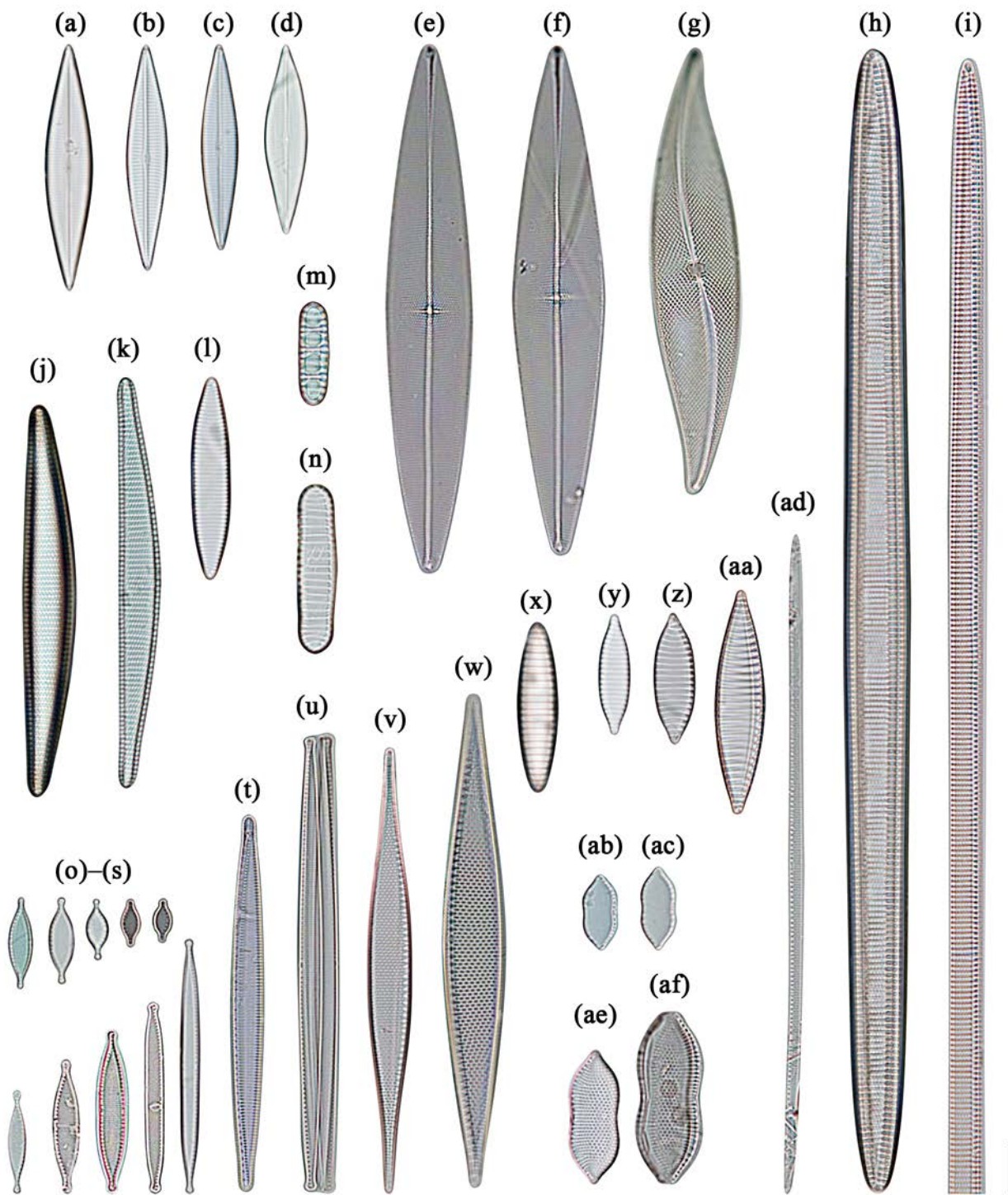

**Figure 10.** (**a**–**d**) *Navicula* cf. *transitantoides*; (**e**,**f**) *Pleurosigma directum*; (**g**) *Pleurosigma diversestriatum*; (**h**,**i**) *Alveus marinus*; (**j**,**k**) *Fragilariopsis doliolus*; (**l**) *Fragilariopsis* aff. *oceanica*; (**m**,**n**) *Neodenticula seminae*; (**o**–**s**) *Nitzschia bicapitata*; (**t**,**u**) *Nitzschia interrruptestriata*; (**v**,**w**) *Nitzschia kolaczeckii*; (**x**) *Nitzschia sicula* var. *sicula*; (**y**–**aa**) *Nitzschia sicula* var. *bicuneata*; (**ab**,**ac**) *Psammodictyon* sp.; (**ad**) *Pseudo-nitzschia turgiduloides*; (**ae**,**af**) *Tryblionella coarctata*. Scale bar = 10 μm.

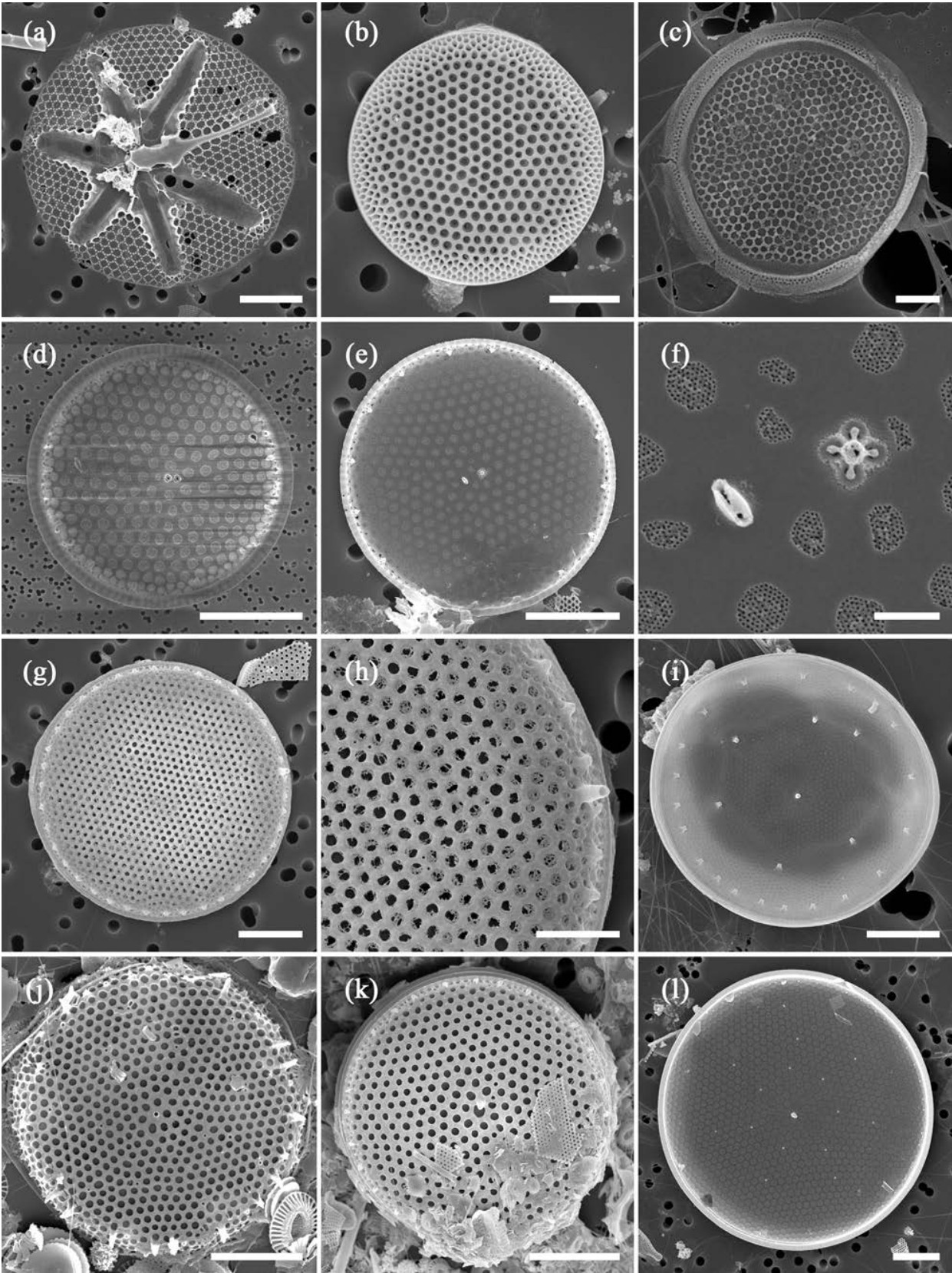

**Figure 11.** (**a**) *Asteromphalus heptactis*; (**b**) *Actinocyclus iraidae*; (**c**) *Minidiscus trioculatus*; (**d**) *Shionodiscus trifultus*; (**e**,**f**) *Shionodiscus variantus*; (**g**,**h**) *Thalassiosira anguste-lineata*; (**i**) *Thalassiosira diporocyclus*; (**j**) *Thalassiosira mendiolana*; (**k**,**l**) *Thalassiosira punctifera*. Scale bars = 10 μm ((**a**,**e**,**g**,**k**,**l**)), 5 μm ((**b**,**d**,**h–j**)), 1 μm ((**c**,**f**)).

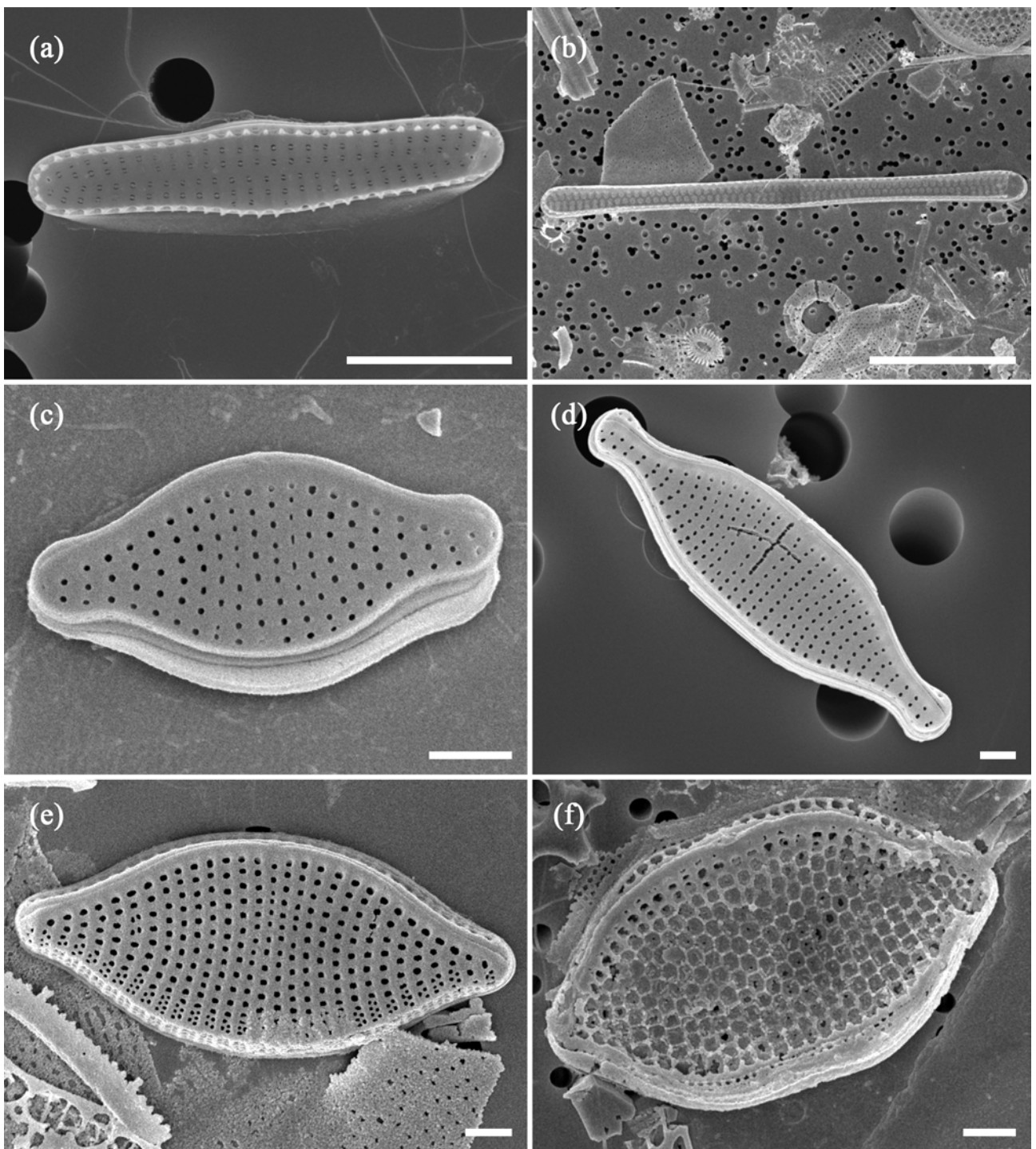

**Figure 12.** (**a**,**b**) *Neodelphineis indica*; (**c**,**d**) *Nitzschi bicapitata*; (**e**) *Nitzschia bifurcata*. (**f**) *Psaamodictyon* sp. Scale bars = 10 μm ((**b**)), 5 μm ((**a**)), 1 μm ((**c**–**f**)).

**Subphylum Coscinodiscophytina Medlin & Kaczmarska, 2004**
**Class Coscinodiscophyceae Round & R.M. Crawford in Round et al., 1990**
**Subclass Coscinodiscophycidae Round & R.M. Crawford in Round et al., 1990**
**Order Asterolamprales Round & R.M. Crawford in Round et al., 1990**
**Family Asterolampraceae H.L. Smith, 1872**
**Genus *Asteromphalus* Ehrenberg, 1844**
***Asteromphalus flabellatus* (Brébisson) Greville, 1859 [Figure 2a (LM)]**

Basionym: *Spatangidium flabellatum* Brébisson, 1857.
References: Hasle and Syvertsen [19], p. 135, pl. 24.
Sample: 2017-KE-02.
Dimensions: Valves 35.7 μm in diameter; twelve areolae in 10 μm, seven hyaline rays.
Diagnostic: Valve pear-shaped; central area slightly eccentric; separating lines straight; hyaline ray slightly tapered toward the valve margin.
Distribution: *Asteromphalus flabellatus* is distributed in a warm-water region [19,20]. In the Pacific Ocean, this species was reported from the central Indo-Pacific to the temperate northern Pacific along the Kuroshio Current: Thailand [21], Indonesia [22], the Philippines, and Yokohama [23]. This species may be transported along the Kuroshio Current.

### *Asteromphalus heptactis* (Brébisson) Ralfs in Pritchard, 1861 [Figure 11a (SEM)]

Basionym: *Spatangidium heptactis* Brébisson, 1857.
References: Hasle and Syvertsen [19], p. 137, pl. 24; Tiffany and Hernández-Becerril [24], p. 254, pl. 2, Fig 2; pl. 15, Figs 1–4; pl. 16, Figs 1–4; pl. 17, Figs 1–6; pl. 18, Figs 1–6; pl. 19, Figs 1–6.
Sample: 2017-KE-18.
Dimensions: Valves 30.8–44.0 μm in diameter; seven areolae in 10 μm, seven hyaline rays.
Diagnostic: Valve circular; central area slightly eccentric; number of areolae between two rays is two to three, except for short rays.
Distribution: *Asteromphalus heptactis* is distributed in a temperate region [19]; this species was also reported from cold-water regions such as the Southern Ocean [25] and Scotch Cap, Alaska [26]. This species may be transported along the Oyashio Current.

### Genus *Spatangidium* Brébisson, 1857
### *Spatangidium arachne* Brébisson, 1857 [Figure 2b (LM)]

Synonym: *Asteromphalus arachne* (Brébisson) Ralfs in Pritchard, 1861.
References: Hernández-Becerril [27], p. 279, Figs 1–14; Tiffany and Hernández-Becerril [24], p. 256, pl. 26, Figs 1–7, pl. 27, Figs 1–6.
Dimensions: Valves 45 μm in diameter; seven to eight areolae in 10 μm.
Diagnostic: Valves oval; central area eccentric and very small; five hyaline rays are narrow, almost straight, except for two short rays that are slightly curved; separating lines are straight; number of areolae between two rays is two to three, except for short rays; three rimoportulae are present near the distal ends of the aborted ray in the central area.
Distribution: *Spatangidium arachne* has been widely reported from tropical to subtropical regions [24]: off the coasts of Baja California; the Gulf of California; the Indian Ocean; Australian waters [27]; and San Diego Bay [24]. Based on the tropical distribution, this species may be transported along the Kuroshio Current.
Remarks: *Spatangidium arachne* was transferred to the genus *Asteromphalus* by Ralfs [28]. However, Hernández-Becerril [27] reinstated it to *Spatangidium* by the presence of one to three rimoportulae near the central area, and the cribra structure that was distinct to *Asteromphalus*.

### Order Coscinodiscales Round & R.M. Crawford in Round et al., 1990
### Family Coscinodiscaceae Kützing, 1844
### Genus *Coscinodiscopsis* E.A. Sar & I. Sunesen, 2008
### *Coscinodiscopsis jonesiana* (Greville) E.A. Sar & I. Sunesen, 2008 [Figure 2c,d (LM)]

Basionym: *Eupodiscus jonesianus* Greville, 1862.
Synonyms: *Coscinodiscus jonesianus* (Greville) Ostenfeld, 1915.
References: Sar et al. [29], p. 419, Figs 1–12.
Dimensions: Valves 55.3–100.4 μm in diameter; areolae in 10 μm, five to seven in center and nine to eleven in margin.
Diagnostic: Valves convex, flat in the center with gently curved margins; areolae fasciculate and form spiral decussate arcs.

Distribution: *Coscinodiscopsis jonesiana* is distributed in warm-water regions [19]. This species may be transported along the Kuroshio Current.

**Genus *Coscinodiscus* Ehrenberg, 1839**
***Coscinodiscus argus* Ehrenberg, 1839 [Figure 2e (LM)]**

References: Hasle and Sims [30], p. 308, Figs 1–7, 33, 34; Hasle and Syvertsen [19], p. 103, pl. 15.

Sample: 2017_KE-8-20.

Dimensions: Valve 58 μm in diameter; areolae in 10 μm, four in center and five in margin.

Diagnostic: Valves flat with rounded margin and steep, high valve mantle; areolae increase from the valve center to the middle of the radius, and then decrease in the margin.

Distribution: *Coscinodiscus argus* was first reported as a fossil from Caltanisetta, Sicily, and Oran, Algeria, and as a living organism from Cuxhaven, North Sea (Ehrenberg [31]). This species has mainly been reported in fossilized materials (e.g., [31,32]). Hasle and Syvertsen [19] mentioned that this species is widely distributed. Therefore, it is difficult to determine which current is related to the occurrence of *C. argus* in the KE region.

***Coscinodiscus centralis* Ehrenberg, 1839 [Figure 2f (LM)]**

References: Hasle and Lange [33], p. 45, Figs 15–30; Hasle and Syvertsen [19], p. 102.
Dimensions: Valves 180.1 μm in diameter; six areolae in 10 μm.

Diagnostic: Valves convex, flat in the center; central rosette distinct and consists of larger-sized central areolae that have visible cribra in LM; areolae decrease in size outward.

Distribution: Although *C. centralis* is widely distributed in the world [19], this species has been reported in warm-water regions: the Philippines [34] and Tanzania [35]. This species may be transported along the Kuroshio Current.

***Coscinodiscus gigas* Ehrenberg, 1841 [Figure 2j (LM)]**

References: Schmidt [36], pl. 64, Fig 1; Takano [37], p. 134, pl. 1, Figs 1–8, pl. 2, Figs 9–16.

Sample: 2017_KE-8-01.

Dimensions: Valve 191.6 μm in diameter; two areolae in 10 μm.

Diagnostic: Valves flat; central rosette absent; areolae adjacent to center, elongated in radial direction and increasing in size outward.

Distribution: *Coscinodiscus gigas* has been reported from warm water regions in the Pacific Ocean: Malaysia [38], Thailand [21], Nicobar Islands of India [36], and Tokyo Bay of Japan [37]. This species may be transported along the Kuroshio Current.

***Coscinodiscus* cf. *marginatus* Ehrenberg, 1843 [Figure 2g,h (LM)]**

References: Sancetta [39], p. 240, pl. 1, Figs 1–13; Hasle and Syvertsen [19], p. 107, pl. 18; Nikolaev et al. [40] p. 17, pl. 16, Figs 1–6.

Sample: 2017_KE-8-01; 2017_KE-8-20.

Dimensions: Valves 21.5–21.7 μm in diameter; areolae in 10 μm, four to five in center and six in margin.

Diagnostic: Valves heavily silicified; areolae coarse.

Distribution: *Coscinodiscus marginatus* is widely distributed in the world [19]. Due to its wide occurrence along the boundary of the northwestern Pacific Ocean, it is difficult to determine which current is related to the occurrence of *C.* cf. *marginatus* in the KE region.

Comments: The valve size of *C. marginatus* has a range of 30 to 200 μm [19], but the materials from the KE were smaller than those in other reports.

***Coscinodiscus radiatus* Ehrenberg, 1840 [Figure 2i (LM)]**

References: Hasle and Syvertsen [19], p. 107, Figs 6d,e, pl. 18; Sar et al. [41], p. 517, Figs 33–50.

Sample: 2017_KE-8-02; 2017_KE-8-06; 2017_KE-8-09; 2017_KE-8-18.

Dimensions: Valves 38.8–53.3 μm in diameter; areolae in 10 μm, four to five in center and five to six in margin.

Diagnostic: Valves flat; areolae coarse.

Distribution: *Coscinodiscus radiatus* is widely distributed in the world [19], from tropical waters such as Thailand [21,42] and the Philippines [34] to cold waters [43]. Due to its wide occurrence along the boundary of the northwestern Pacific Ocean, it is difficult to determine which current is related to the occurrence of *C. radiatus* in the KE region.

**Family Heliopeltaceae H.L. Smith, 1872**
**Genus *Actinoptychus* Ehrenberg, 1843**
***Actinoptychus senarius* (Ehrenberg) Ehrenberg, 1843 [Figure 2l,m (LM)]**

References: Lee and Chang [44], p. 367, Figs 7–9, 12; Siqueiros-Beltrones and Argumedo-Hernández [45], Fig 19; Siqueiros-Beltrones et al. [46], Fig. 24; Uematsu et al. [47], Fig 3; López-Fuerte et al. [48], p. 7, Figs 4a–j.

Dimensions: Valves 39.8 μm in diameter.

Sample: 2017_KE-8-01.

Distribution: In the North Pacific Ocean, *A. senarius* was reported from warm temperate regions: Jinhae Bay of South Korea [44], Tokyo Bay of Japan [47], and the west coast of Mexico [45,46,48]. The occurrence of *A. senarius* was found in the KE region, and it was transported along the Kuroshio Current.

**Family Hemidiscaceae Hendey ex Hasle, 1996**
**Genus *Actinocyclus* Ehrenberg, 1837**
***Actinocyclus curvatulus* Janisch, 1878 [Figure 3c (LM)]**

References: Hasle and Syvertsen [19], p. 121, pl. 19.

Sample: 2017_KE-8-01.

Dimensions: Valves 22.8 μm in diameter; nine areolae in 10 μm.

Diagnostic: Areolae row curved.

Distribution: *Actinocyclus curvatulus* is a cosmopolitan species, but is common to the Arctic Ocean [19]. The occurrence of *A. curvatulus* has been found in the KE region; it was transported along the Oyashio Current.

***Actinocyclus octonarius* Ehrenberg, 1837 [Figure 3a (LM)]**

References: Hasle and Syvertsen [19], p. 120.

Sample: 2017_KE-8-01.

Dimensions: Valves 52.9 μm in diameter; ten areolae in 10 μm.

Distribution: *Actinocyclus octonarius* is a cosmopolitan species [19]. Due to its wide occurrence along the boundary of the northwestern Pacific Ocean, it is difficult to determine which current is related to the occurrence of *A. octonarius* in the KE region.

***Actinocyclus ochotensis* Jousé, 1968 [Figure 3b (LM)]**

References: Koizumi [49], p. 831, pl. 2, Figs 3–7.

Sample: 2017_KE-8-01.

Dimensions: Valves 37.9 μm in diameter; eight areolae in 10 μm.

Diagnostic: Valve center with a hyaline area without areolae; areolae sparse in the valve center and packed toward the margin; areolation fasciculate; pseudonodule distinct in the valve mantle.

Distribution: *Actinocyclus ochotensis* was first described in sediment from the Okhotsk Sea [50]; since then, there have been additional records from sediments around this Sea [49,51]. The occurrence of *A. ochotensis* has been found in the KE region; it was transported along the Oyashio Current.

***Actinocyclus iraidae* Gogorev, 2015 [Figure 3d,e (LM), Figure 11b (SEM)]**

Synonyms: *Thalassiosira variabilis* Makarova, 1959; *Actinocyclus variabilis* (Makarova) Makarova, 1985.

References: Makarova [52], p. 85, Figs 7–11; Gogorev and Pushina [53].

Sample: 2017_KE-8-02; 2017_KE-8-09; 2017_KE-8-18.

Dimensions: Valves 12.3–24.5 μm in diameter; eight to twelve areolae in 10 μm.

Diagnostic: Areola row eccentric; a single isolated areola in the center; pseudonodule uncertain in LM.

Distribution: *Actinocyclus iraidae* was regarded as a brackish to euryhaline and eury-thermal neritic [54]. Since the original description, there have been no additional records from other regions. Due to the lack of additional records along the boundary of the north-western Pacific Ocean, it is difficult to determine which current is related to the occurrence of *A. iraidae* in the KE region.

Comments: Makarova [52] first described this species in the Caspian Sea as *Thalassiosira variablis*, and she changed it to *Actinocycylus* by means of SEM examination of the same material [54]. Subsequently, Gogorev and Pushina [53] recognized that *A variabilis* (Makarova) Makarova is a homonym of *A. variabilis* Corda, so they renamed it *A. iraidae*.

**Genus *Azpeitia* M. Peragallo, 1912**
***Azpeitia neocrenulata* (VanLandingham) Fryxell & Watkins, 1986 [Figure 3f,g (LM)]**

References: Fryxell et al. [55], p. 18, Figs 16.1–16.3D, Fig 30.2; Garcia and Odebrecht [56], p. 422, Figs 24–26.

Sample: 2017_KE-8-01.

Dimensions: Valves 19–44.7 μm in diameter; six to ten areolae in 10 μm.

Diagnostic: Areolae rows fasciculate with marginal rimoportula and a depression at the end of each fascicle.

Distribution: Fryxell et al., [55] mentioned that *A. neocrenulata* has been found in the Gulf of Mexico, the central Pacific, and the Indian Ocean as well as in the Gulf Stream Warm Core Ring of the North Atlantic as a warm-water species; Garcia and Odebrecht [56] observed it in Brazilian water. This species may be transported along the Kuroshio Current.

***Azpeitia nodulifera* (Schmidt) Fryxell & P.A. Sims [Figure 3h,i (LM)]**

References: Fryxell, Sims and Watkins [55], p. 19, Figs 17, 18.1–18.5, 30.3, 30.4; Garcia and Odebrecht [56], p. 422, Figs 22, 23, 31–33.

Sample: 2017_KE-8-02.

Dimensions: Valve 27 μm in diameter; seven areolae in 10 μm.

Diagnostic: Large rimoportula close to valve center.

Distribution: Fryxell, Sims and Watkins [55] mentioned that *A. nodulifera* has been found in the central Pacific, Gulf of Mexico, and the northwest Atlantic Ocean in Gulf Stream warm core rings; Garcia and Odebrecht [56] observed it in Brazilian water. This species may be transported along the Kuroshio Current.

**Genus *Roperia* Grunow ex Pelletan, 1889**
***Roperia tesselata* (Roper) Grunow ex Pelletan, 1889 [Figure 3j,k (LM)]**

Basionym: *Eupodiscus tesselatus* Roper, 1858.

References: Lee and Lee [57], p. 327, Figs 1–19.

Sample: 2017_KE-8-02; 2017_KE-8-06; 2017_KE-8-20.

Dimensions: Valve 23–63.9 μm in diameter; four to eight areolae in 10 μm.

Distribution: Hasle [58] regarded *Roperia tesselata* as a warm-water species. This species may be transported along the Kuroshio Current.

Comments: Valves variable from circular to tear-drop form [57].

**Order Stellarimales Nikolaev & Harwood in Witkowski & Siemińska, 2000**
**Family Stellarimaceae Nikolaev ex P.A. Sims & Hasle, 1990**
**Genus *Stellarima* Hasle & P.A. Sims, 1986**
***Stellarima stellaris* (Roper) Hasle & P.A. Sims, 1986 [Figure 2k (LM)]**

Basionym: *Coscinodiscus stellaris* Roper, 1858.

Synonym: *Symbolophora stellaris* (Roper) Nikolaev, 1983.

References: Hasle et al. [59], p. 198, Figs 26–28.

Dimensions: Valves 73.3 μm in diameter; eleven areolae in 10 μm.

Diagnostic: Valves extremely convex; areolation arranged in wide sectors; central area is filled with areolae; number of rimoportula is three to four.

Distribution: Hasle and Syvertsen [19] regarded this species as a warm-water to temperate planktonic species. This species may be transported along the Kuroshio Current.

Comments: *Stellarima stellaris* is distinguished from *S. microtrias* by the smaller areolae, wider sectors, a narrow hyaline margin, and a valve center almost filled by areolae [59].

**Subphylum Probosciophytina D.G. Mann in Adl et al., 2019**
**Order Probosciales Medlin, 2021**
**Family Probosciaceae R.W. Jordan & Ligowski, 2004**
**Genus *Proboscia* B.G. Sundström, 1986**
***Proboscia indica* (H. Peragallo) Hernández-Becerril, 1995 [Figure 4a (LM)]**

References: Hernández-Becerril [60], p. 254, Figs 5, 6; Yun and Lee [61], p. 304, Figs 2A–F; Boonprakob et al. [62], p. 170, Figs 189–198.

Sample: 2017_KE-8-02.

Distribution: *Proboscia indica* is distributed in temperate to tropical waters [60]. In the North Pacific Ocean, this species was reported in Thailand [62] and South Korea [61]. Based on the warm-water occurrence of this species, it may be transported along the Kuroshio Current.

**Subphylum Rhizosoleniophytina D.G. Mann in Adl et al., 2019**
**Subclass Rhizosoleniophycidae Round & Crawford in Round et al., 1990**
**Order Rhizosoleniales P.C. Silva, 1962**
**Family Rhizosoleniaceae De Toni, 1890**
**Genus *Pseudosolenia* B.G, Sundström, 1986**
***Pseudosolenia calcar-avis* (Schultze) B.G. Sundström, 1986 [Figure 4b (LM)]**

References: Hasle and Syvertsen [19], p. 160, pl. 30; Yun and Lee [61], p. 307, Figs 4A–H.

Sample: 2017_KE-8-02; 2017_KE-8-04.

Distribution: *Pseudosolenia calcar-avis* is a warm-water species [19]. This species may be transported along the Kuroshio Current.

**Genus *Rhizosolenia* Brightwell, 1858**
***Rhizosolenia bergonii* H. Peragallo, 1892 [Figure 4c–e (LM)]**

References: Teanpisut and Patarajinda [21], Figs 2–17; Yun and Lee [63], p. 176, Figure 2A–G; Boonprakob et al. [62], p. 153, Figs 18–34.

Sample: 2017_KE-8-01; 2017_KE-8-02; 2017_KE-8-09.

Distribution: *Rhizosolenia bergonii* is distributed in warm-water regions [19]: Thailand [21,62], Japan [64], and South Korea [63]. Based on the warm water of occurrence of this species, it may be transported along the Kuroshio Current.

***Rhizosolenia hebetata* f. *semispina* (Hensen) Gran, 1908 [Figure 4f,g (LM)]**

References: Sundström [65], p. 48, Figs 19, 117; Hasle and Syvertsen [19], p. 149.
Sample: 2017_KE-8-01, 2017_KE-8-05.

Distribution: *Rhizosolenia hebetata* f. *semispina* is distributed in northern cold-water regions [19,65]. This species may be transported along the Oyashio Current.

***Rhizosolenia imbricata* Brightwell 1858 [Figure 4h (LM)]**

References: Sundström [65], p. 80, Figs 200–208, Hasle and Syvertsen [19], p. 155.
Sample: 2017_KE-8-04.

Distribution: *Rhizosolenia imbricata* is distributed worldwide, except in polar regions [19,65]. This species may be transported along the Kuroshio Current.

*Rhizosolenia styliformis* **Brightwell, 1858 [Figure 4i (LM)]**

References: Cleve [22], p. 11, Ostenfeld [42], p. 13; Sundström [65], p. 15, Figs 5, 47–56; Hasle and Syvertsen [19], p. 146; Boonprakob et al. [62], p. 151, Figs 3–8.
Sample: 2017_KE-8-01.
Dimensions: 14 μm in diameter.
Distribution: *Rhizosolenia styliformis* is distributed in the northern part of the Atlantic [19,65], but has also been found in tropical waters such as the Gulf of Siam in Thailand [42,62] and the Java Sea [22]. Due to its occurrence in a wide range of temperatures, it is difficult to determine which current is related to the occurrence of *C. radiatus* in the KE region.

**Genus *Sundstroemia* Medlin, Boonprakob, Lundholm & Moestrup, 2021**
***Sundstroemia pungens* (Cleve-Euler) Medlin, Lundholm, Boonprakob & Moestrup, [Figure 4j (LM)]**

References: Yun et al. [66], p. 142, Figs 1A–F; Boonprakob, Lundholm, Medlin and Moestrup [62], p. 162, Figs 106–118 (as *Rhizosolenia* aff. *pungens*).
Sample: 2017_KE-8-01.
Distribution: *Sundstroemia pungens* has been found in warm to temperate regions: Thailand [62] and the East China Sea of South Korea [66]. This species may be transported along the Kuroshio Current.

**Subphylum Bacillariophytina Medlin & Kaczmarska, 2004**
**Class Mediophyceae Medlin & Kaczmarska, 2004**
**Subclass Thalassiosirophycidae Round & R.M. Crawford in Round et al., 1990**
**Order Thalassiosirales Glezer & Makarova, 1986**
**Family Lauderiaceae (Schütt) Lemmermann, 1899**
**Genus *Lauderia* Cleve, 1873**
***Lauderia annulata* Cleve, 1873 [Figure 5a (LM)]**

References: Hasle [67], p. 3, Figs 1–3; Hasle and Syvertsen [19], p. 36.
Sample: 2017_KE-8-09.
Dimensions: Valves 58.6 μm in diameter.
Diagnostic: Valve face with faint radial ribs; central annulus distinct; numerous fultoportulae scattered on valve face and margin.
Distribution: *Lauderia annulata* is distributed in temperate to warm-water regions [19]. This species may be transported along the Kuroshio Current.

**Family Thalassiosiraceae Lebour, 1930**
**Genus *Detonula* F.Schütt ex De Toni, 1894**
***Detonula confervacea* (Cleve) Gran, 1900 [Figure 5b,c (LM)]**

Basionym: *Lauderia confervacea* Cleve, 1896.
Synonym: *Detonula cystifera* Gran, 1900.
References: Hasle [67], p. 15, Figs 44–68; Syvertsen [68], p. 55, Figs 63–69; Hasle and Syvertsen [19], p. 36.
Sample: 2017_KE-8-01; 2017_KE-8-05.
Dimensions: Valves 18.5–21.1 μm in diameter.
Diagnostic: Valve face with faint radial ribs; single central fultoportula; one ring of marginal fultoportulae with distinct external tube.
Distribution: *Detonula confervacea* is distributed in northern cold to northern temperate regions [19]. This species may be transported along the Oyashio Current.

**Genus *Minidiscus* Hasle 1973**
***Minidiscus trioculatus* (F.J.R. Taylor) Hasle 1973 [Figure 11c (SEM)]**

Basionym: *Coscinodiscus trioculatus* F.J.R. Taylor 1966.
References: Hasle [67], p. 29, Figs 101–108; Kaczmarska et al. [69], p. 464, Figs 7–10.
Sample: 2017_KE-8-06.

Dimensions: Valves 4.1–5.4 μm in diameter; 37–45 areolae in 10 μm.

Diagnostic: Prominent hyaline in valve margin; hexagonal areolae; three to four valve face fultoportulae; rimoportulae on valve face.

Distribution: *Minidiscus trioculatus* is a cosmopolitan species [67,69]. Due to its occurrence in a wide range of temperatures, it is difficult to determine which current is related to the occurrence of *M. troiculatus* in the KE region.

**Genus *Planktoniella* F.Schütt, 1892**
***Planktoniella blanda* (A. Schmidt) Syvertsen & Hasle, 1993 [Figure 5d (LM)]**

Basionym: *Coscinodiscus blandus* A. Schmidt, 1878.

Synonyms: *Coscinodiscus bipartitus* Rattray, 1890; *Coscinodiscus latimarginatus* Guo, 1981; *Thalassiosira blandus* (A. Schmidt) Desikachary & Gowthaman in Desikachary, 1989; *Thalassiosira bipartita* (Rattray) Hallegraeff, 1992.

References: Hasle and Syvertsen [70], p. 304, Figs 19–31; Hasle and Syvertsen [19], p. 40.

Sample: 2017_KE-8-01; 2017_KE-8-02.

Dimensions: Valves 34 μm in diameter; five to seven areolae in 10 μm.

Diagnostic: Prominent hexagonal areolae arranged in linear formation. Marginal ribs distinct. Single central fultoportulae close to central areola.

Distribution: *Planktoniella blanda* is distributed in temperate to warm-water regions [19]. This species may be transported along the Kuroshio Current.

***Planktoniella sol* (G.C.Wallich) Schütt, 1892 [Figure 5e (LM)]**

References: Hasle and Syvertsen [70], p. 303, Figs 17–31; Hasle and Syvertsen [19], p. 39.

Sample: 2017_KE-8-01.

Dimensions: Valves 33.5 μm in diameter; six to eight areolae in 10 μm.

Diagnostic: Areolae arranged in eccentric pattern. Marginal ribs distinct. Single central fultoportula close to central areola. Two rimoportulae in the valve margin.

Distribution: *Planktoniella sol* is distributed in warm-water regions [19]. This species may be transported along the Kuroshio Current.

**Genus *Shionodiscus* A.J. Alverson, S.H. Kang & E.C. Theriot, 2006**
***Shionodiscus oestrupii* var. *oestrupii* (Ostenfeld) A.J. Alverson, S.H. Kang & E.C. Theriot, 2006 [Figure 5f–j (LM)]**

Basionym: *Cosinosira oestrupii* Ostenfeld, 1900.

Synonym: *Thalassiosira oestrupii* (Ostenfeld) Proshkina-Lavrenko ex Hasle, 1960.

References: Fryxell and Hasle [71], p. 805, Figs 1–10; Hasle and Syvertsen [19], p. 83 (as *Thalassiosira oestrupii*).

Sample: 2017_KE-8-01; 2017_KE-8-02; 2017_KE-8-05; 2017_KE-8-06; 2017_KE-8-20.

Dimensions: Valves 14.1–20.5 μm in diameter; five to seven areolae in 10 μm.

Diagnostic: Areolae larger in valve center than marginal one. Marginal fultoportula dense.

Distribution: *Shionodiscus oestrupii* is distributed in temperate to warm-water regions [19]. This species may be transported along the Kuroshio Current.

***Shionodiscus* cf. *oestrupii* var. *venrickae* (Fryxell & Hasle) A.J. Alverson, S.H. Kang & E.C. Theriot, 2006 [Figure 5k (LM)]**

References: Lee and Yoo [72], p. 190, pl. 4, Figs 1–5; Hernández-Becerril and Peña [73], p. 550, Figs 42, 43; Aké-Castillo et al. [74], p. 495, Fig 21; Naya [75], p. 157, Figs 25–26, 65–68.

Sample: 2017_KE-8-01; 2017_KE-8-02; 2017_KE-8-05; 2017_KE-8-06; 2017_KE-8-20.

Dimensions: Valves 15.7–54.3 μm in diameter; six to ten areolae in 10 μm.

Diagnostic: Areolation distinct and eccentric.

Distribution: *Shionodiscus oestrupii* var. *venrickae* is distributed in temperate to warm-water regions of the Pacific Ocean: Gulf of California [73]; Gulf of Tehuantepec of Mexico [74]; Yellow Sea of South Korea [72]; and central Kanto Plain of Japan [75]. This species may be transported along the Kuroshio Current.

Comments: The *Shionodiscus ostrupii* complex is characterized by the single central fultoportula and valve face rimoportula away from the central fultoportula by two to three areolae. The KE specimen differs in valve size from *S. oestrupii* var. *venrickae*, which has 5.5–39 μm valves [71].

### *Shionodiscus trifultus* (Fryxell) A.J. Alverson, S.H. Kang & E.C. Theriot, 2006 [Figure 11d (SEM)]

Basionym: *Thalassiosira trifulata* Fryxell, 1979.

References: Fryxell and Hasle [76], p. 16, Figs 1–24; Hasle and Syvertsen [19], p. 87, pl. 12 (as *Thalassiosira trifulta*).

Sample: 2017_KE-8-01; 2017_KE-8-02; 2017_KE-8-05; 2017_KE-8-06; 2017_KE-8-20.

Dimensions: Valves 19.9 μm in diameter; five to seven areolae in 10 μm.

Diagnostic: Areolation eccentric; rimoportula close to valve margin; fultoportula with internally trifultate structure; one to three fultoportulae in valve center.

Distribution: *Shionodiscus trifultus* is distributed in temperate to cold-water regions [19]. This species may be transported along the Oyashio Current.

### *Shionodiscus variantus* (Shiono) A.J. Alverson, S.H. Kang & E.C. Theriot, 2006 [Figure 11e,f (SEM)]

Basioinym: *Thalassiosira variantia* Shiono, 2001.

References: Shiono [77], p. 86, Figs 13–24.

Sample: 2017_KE-8-09; 2017_KE-8-18; 2017_KE-8-20.

Dimensions: Valves 25.1–30.0 μm in diameter; seven to eight areolae in 10 μm.

Diagnostic: Areolation eccentric; rimoportula close to central fultoportula; fultoportula with internally operculate structure, larger in valve center than marginal one; marginal fultoportula dense.

Distribution: *Shionodiscus variantus* was originally described in the late Pliocene sediments of the Northwest Pacific Ocean [77]. Although there are no reports on *S. variantus* from other regions since this species was originally found in a fossil deposit, it seems to be overlooked. This species may be transported along the Oyashio Current.

### *Shionodiscus* aff. *poroirregulatus* (Hasle & Heimdal) A.J. Alverson, S.H. Kang & E.C. Theriot, 2006 [Figure 5l (LM)]

Basionym: *Thalassiosira poroirregulata* Hasle & Heimdal, 1970.

References: Rivera [78], p. 113, pl. 51, Figs 317–322, pl. 52, Figs 323–328, pl. 53, Figs 329–334, pl. 54, Figs 335–339; Lee and Park [79], p. 554, Figs 28–29.

Sample: 2017_KE-8-09; 2017_KE-8-20.

Dimensions: Valves 18.3–41.3 μm in diameter; five to eight areolae in 10 μm.

Diagnostic: Areola arranged eccentrically; central and subcentral fultoportulae variable.

Comments: The distinguishing characteristics of *Shionodiscus poroirregulatus* are irregular subcentral fultoportulae and fasciculate areolation [19]. The KE specimens also have an irregular number of subcentral fultoportulae, but they differ in eccentric areolation and coarse areolae. Since *S. poroirregulatus* was originally described in Chilean waters, this species has mainly been reported from the southern hemisphere, including the Antarctic to the Subantarctic Ocean [80]. Although Lee and Park [79] reported *S. poroirregulatus* in South Korean waters, it also has coarser areolae than the Chilean *S. poroirregulatus*. The taxonomic status of *Shionodiscus* aff. *proroirregulatus* in the northern hemisphere needs additional research.

**Genus *Takanoa* I.V. Makarova, 1994**
***Takanoa bingensis* (Takano) I.V. Makarova, 1994 [Figure 5m (LM)]**

Basionym: *Thalassiosira bingensis* Takano, 1980.
References: Takano [81], p. 42, Figs 1–34; Makarova [82], p. 105.
Sample: 2017_KE-8-01.
Dimensions: Valves 56.2 μm in diameter; areolae invisible in LM.
Diagnostic: Valves weak; areolae invisible in LM; marginal fultoportulae dense.
Distribution: *Takanoa bingensis* was first observed in the coastal waters of Japan (Takano 1980). Since then, there has been no record of it in other regions. Due to the lack of additional occurrence of this species, it is difficult to determine which current is related to the occurrence of *T. bingensis* in the KE.

Comments: Makarova [82] separated this species from the genus *Thalassiosira* by consideration of its characteristics such as the two rings of the marginal and mantle fultoportulae, two rimoportulae surrounded by small arcs formed by fultoportulae, and an unusual cingulum.

**Genus *Thalassiosira* Cleve, 1873**
***Thalassiosira anguste-lineata* (A. Schmidt) Fryxell & Hasle, 1977 [Figure 11g,h (SEM)]**

Basionym: *Coscinodiscus anguste-lineata* A. Schmidt, 1878.
Synyonms: *Coscinodiscus polychordus* Gran 1897; *Coscinodiscus lineata* f. *polychordus* (Gran) H. Peragallo & M. Peragallo, 1908; *Coscinodiscus lineatus* var. *polychorda* (Gran) F.W. Mills, 1933; *Coscinosira polychorda* (Gran), 1900; *Thalassiosira polychorda* (Gran) E. Jørgensen, 1899.
References: Fryxell and Hasle [83], p. 73, pl. 4, 5, Figs 22–26, 27–34; Hasle and Syvertsen 1996, p. 71, pl. 9; Park et al. [84], p. 49, Fig 6.
Sample: 2017_KE-8-18.
Dimensions: Valves 40.4 μm in diameter; areolae invisible in LM.
Diagnostic: Central fultoportula absent; modified ring of five to seven fultoportulae equidistant from the valve center.
Distribution: *Thalassiosira anguste-lineata* is a cosmopolitan species [19]. Park et al. [84] mentioned that this species is distributed in cold to tropical waters. Due to its occurrence in a wide range of temperatures, it is difficult to determine which current is related to the occurrence of *T. anguste-lineata* in the KE region.

***Thalassiosira diporocyclus* Hasle, 1972 [Figure 5n,o (LM), Figure 11i (SEM)]**

References: Hasle [85], p. 113, Figs 25–45; Miyahara et al. [86], Figs 2A–D; Li et al. [87], p. 91, Figs 36–42; Park et al. [84], p. 408, Fig 12.
Sample: 2017_KE-8-06; 2017_KE-8-09.
Dimensions: Valve 13.4–14.8 μm in diameter.
Diagnostic: Valves flat to convex with curved mantle; areolae invisible in LM; two rings of marginal fultoportulae; a single rimoportula located between two marginal fultoportulae.
Distribution: *Thalassiosira diporocyclus* is distributed in temperate to tropical regions [19]. In the Pacific Ocean, *T. diporocyclus* was found in Japanese, Chinese, and Korean waters [84,86,87]. This species may be transported along the Kuroshio Current.

***Thalassiosira ferelineata* Hasle & Fryxell, 1977 [Figure 5p,q (LM)]**

References: Hasle and Fryxell [88], p. 26, Figs 46–53; Hasle and Syvertsen [19], p 59.
Sample: 2017_KE-8-01; 2017_KE-8-05.
Dimensions: Valve 26.8 μm in diameter; seven areolae in 10 μm.
Diagnostic: Areolation linear; a single central fultoportula close to the central areola.
Distribution: *Thalassiosira ferelineata* is distributed in warm-water regions [19]. This species may be transported along the Kuroshio Current.
Comments: *Thalassiosira ferelineata* can be confused with *T. tenera* due to the linear areolation, but they are different regarding the position of the central fultoportula. The

former has a central fultoportula close to the central areolae between areolae, but in the latter, the central fultoportula replaces the central areola.

### *Thalassiosira lineata* Jousé, 1968 [Figure 5r–t (LM)]

References: Hasle and Fryxell [88], p. 22, Figs. 15–25; Hasle and Syvertsen [19], p 80; Park et al. [84], p. 411, Fig 19.
Sample: 2017_KE-8-01; 2017_KE-8-05; 2017_KE-8-06; 2017_KE-8-09.
Dimensions: Valves 17.4–43.9 in diameter, nine to fifteen areolae in 10 μm.
Diagnostic: Areolation linear; valve face has scattered fultoportulae prominent in LM.
Distribution: *Thalassiosira lineata* is distributed in warm-water regions [19]. This species may be transported along the Kuroshio Current.

### *Thalassiosira mendiolana* Hasle & Heimdal, 1970 [Figure 11j (SEM)]

References: Hasle and Heimdal [89], p. 570, Figs 44–53, 73 and 74; Fryxell and Hasle [90], Figs 22–32; Park et al. [84], p. 413, Fig 24.
Sample: 2017_KE-8-05
Dimensions: Valves 16.6 in diameter; 17 areolae in 10 μm.
Diagnostic: Areolae fasciculate; numerous fultoportulae on valve face; marginal spine.
Distribution: *Thalassiosira mendiolana* has been mainly reported in the southern Pacific Ocean [19]. Park, Jung, Lee, Yun and Lee [84] first reported it in the Heuksando Island of the Yellow Sea of South Korea in the northwest Pacific. However, there are few records on this species, which mainly occurs in warm-water regions: Peru and Chilean waters ([78,89]; Gulf of California [73]. This species may be transported along the Kuroshio Current.

### *Thalassiosira punctifera* (Grunow) Fryxell, Simonsen & Hasle, 1974 [Figure 11k,l (SEM)]

Basionym: Coscinodiscus excentricus var. punctifera Grunow, 1884.
References: Simonsen [20], p. 10, pl. 2, Fig 4, pl. 3; Park, Jung, Lee, Yun and Lee [84], p. 417, Fig 36.
Sample: 2017_KE-8-05; 2017_KE-8-06.
Dimensions: Valves 30.3–59.7 in diameter; six to eight areolae in 10 μm.
Diagnostic: Areolation fasciculate; two rimoportulae, one close to the central occluded areola and the other between the marginal fultoportulae.
Distribution: *Thalassiosira punctifera* is distributed in tropical to subtropical regions [84]. This species may be transported along the Kuroshio Current.

### *Thalassiosira sacketii* Fryxell, 1977 [Figure 6a–d (LM)]

References: Fryxell and Hasle [83], p. 76, pl. 6, Figs 35–39; Herzig and Fryxell [91], p. 19; Licea [92], p. 321, pl. 4, Figs 41, 42.
Sample: 2017_KE-8-02; 2017_KE-8-05.
Dimensions: Valves 32.4–39.7 in diameter; nine to thirteen areolae in 10 μm.
Diagnostic: Areolation radial, fasciculate; small spinules scattered on valve face.
Distribution: *Thalassiosira sacketii* has been observed in the Gulf of Mexico [83], Gulf Stream Warm Core Ring [91] in the Atlantic Ocean, and also the Pacific Ocean [92]. This species may be transported along the Kuroshio Current.

### *Thalassisoira subtilis* (Ostenfeld) Gran, 1900 [Figure 6g (LM)]

References: Hasle [85], p. 112, Figs 1–20, 66 and 67; Hasle and Syvertsen [19], p. 58.
Sample: 2017_KE-8-01.
Dimensions: Valve 47.3 μm in diameter; 16 areolae in 10 μm.
Diagnostic: Fultoportulae scattered on valve face; rimoportula somewhat far from valve margin.
Distribution: *Thalassiosira subtilis* is distributed in temperate to warm-water regions [19]. This species may be transported along the Kuroshio Current.

*Thalassiosira symmetrica* **Fryxell & Hasle, 1973 [Figure 6e,f (LM)]**

References: Fryxell and Hasle [90], p. 312, Figs 37-46; Simonsen [20], p. 11, pl. 6, Figs 1, 2; Rivera [78], p. 144; pl. 68, Figs 425–427, pl. 69, Figs 428–431; Herzig and Fryxell [91], p. 22; Hernández-Becerril and Peña [73], p. 552, Figs 59–61, 65; Aké-Castillo et al. [74], p. 497, Figs 33–36.

Sample: 2017_KE-8-01; 2017_KE-8-02; 2017_KE-8-04.

Dimensions: Valves 40.8–61.8 in diameter; five to six areolae in 10 μm.

Diagnostic: Absence of central fultoportula; single central areola surrounded by seven areolae in a symmetrical rosette; central ring of seven fultoportulae separated by two (ranging from one to three) areolae from central areola.

Distribution: *Thalassisoira symmetrica* is distributed from temperate to warm water in oceanic regions [84]. This species may be transported along the Kuroshio Current.

*Thalassiosira tenera* **Proschkina-Lavrenko, 1961 [Figure 6h (LM)]**

References: Hasle and Fryxell [88], p. 28, Figs 54–65; Hasle and Syvertsen [19], p. 59.

Sample: 2017_KE-8-04; 2017_KE-8-06; 2017_KE-8-18.

Dimensions: Valves 8.5–23.2 in diameter; 10–15 areolae in 10 μm.

Diagnostic: Areolation linear; a single central fultoportula replaces central areola; wedge-shaped external tube of marginal fultoportulae.

Distribution: *Thalassiosira tenera* usually occurs in the coastal area in temperate to warm-water regions [84,88]. This species may be transported along the Kuroshio Current.

**Subclass Lithodesmiophycidae Round & R.M. Crawford in Round et al., 1990**
**Order Lithodesmiales Round & R.M. Crawford, 1990**
**Family Lithodesmiaceae Round in Round et al., 1990**
**Genus *Lithodesmium* Ehrenberg, 1839**
*Lithodesmium variabile* **Takano, 1979 [Figure 6i (LM)]**

References: Takano [93], p. 36, Figs 1A–M, 2–21.

Sample: 2017_KE-8-06.

Dimensions: 19.3 μm in diameter.

Diagnostic: Valves irregularly triangular; marginal ridge sparse.

Distribution: *Lithodesmium variabile* was first recorded in the Japanese coastal waters [93], and Hasle personally observed it in the Gulf of Naples [19]. Due to the lack of additional occurrence of this species, it is difficult to determine which current is related to the occurrence of *L. variabile* in the KE region.

**Subclass Chaetocerotophycidae Round & R.M. Crawford in Round et al., 1990**
**Order Hemiaulales Round & R.M. Crawford in Round et al., 1990**
**Family Hemiaulaceae Heiberg 1863**
**Genus *Cerataulina* H. Peragallo ex F. Schütt, 1896**
*Cerataulina pelagica* **(Cleve) Hendey, 1937 [Figure 6j (LM)]**

References: Hasle and Syvertsen [19], p. 171.

Sample: 2017_KE-8-01.

Dimensions: 24 μm in diameter.

Diagnostic: Wing-like elevation less conspicuous, rimoportula central or subcentral.

Distribution: *Cerataulina pelagica* is distributed worldwide [19] but is common to warm-water regions: Indian Ocean [20] and Thailand [21]. This species may be transported along the Kuroshio Current.

**Genus *Hemiaulus* Heiberg, 1863**
*Hemiaulus sinensis* **Greville, 1865 [Figure 6k,l (LM)]**

References: Hasle and Syvertsen [19], p. 177.

Sample: 2017_KE-8-01; 2017_KE-8-02.

Dimensions: Valve 25.8–41.3 μm in apical axis; seven areolae in 10 μm.

Diagnostic: Valve face slightly concave; areolae coarse; elevation long; rimoportula submarginal.

Distribution: *Hemiaulus sinensis* is distributed in temperate to warm-water regions [19]. This species may be transported along the Kuroshio Current.

**Genus *Eucampia* Ehrenberg, 1839**
***Eucampia cornuta* (Cleve) Grunow, 1883 [Figure 6m (LM)]**

References: Hasle and Syvertsen [19], p. 175.
Sample: 2017_KE-8-02.
Dimensions: Valve 45.3 µm in apical axis.
Diagnostic: Valve face concave; elevation long, narrow; rimoportula in a depressed valve center.

Distribution: *Eucampia cornuta* is a warm-water species [19]. This species may be transported along the Kuroshio Current.

**Order Chaetocerotales Round & R.M. Crawford in Round et al., 1990**
**Family Chaetocerotaceae Ralfs in Prichard, 1861**
**Genus *Bacteriastrum* Shabdolt, 1853**
***Bacteriastrum elongatum* Cleve, 1897 [Figure 7a,b (LM)]**

References: Hasle and Syvertsen [19], p. 189, pl. 37.
Sample: 2017_KE-8-01; 2017_KE-8-02.
Distribution: *Bacteriastrum elongatum* is distributed in temperate to warm-water regions [19]; this species may therefore be transported along the Kuroshio Current.

***Bacteriastrum furcatum* Shabdolt, 1853 [Figure 7c–e (LM)]**

References: Hasle and Syvertsen [19], p. 189, pl. 37; Piredda et al. [94], Fig 2; Park et al. [95], p. 9, Fig 81.
Sample: 2017_KE-8-01; 2017_KE-8-02; 2017_KE-8-09.
Distribution: *Bacteriastrum furcatum* is distributed in temperate to warm-water regions [94]; this species may therefore be transported along the Kuroshio Current.

**Genus *Chaetoceros* Ehrenberg, 1844**
***Chaetoceros affinis* Lauder, 1864 [Figure 8a (LM)]**

References: Hasle and Syvertsen [19], p. 217, pl. 46; Lee et al. [96], p. 615, Figs 1, 45–48.
Sample: 2017_KE-8-01.
Diagnostic: Terminal setae large, strongly divergent.
Distribution: *Chaetoceros affinis* is a cosmopolitan species [19], but is frequently found in the tropical regions of the west Pacific: the Philippines [34]; Thailand [21]; Hong Kong [97]; South Korea [96]. This species may be transported along the Kuroshio Current.

***Chaetoceros atlanticus* Cleve, 1873 [Figure 8b (LM)]**

References: Cupp [98], p. 103, Fig 59A; Hasle and Syvertsen [19], p. 196; Lee et al. [99], p. 175, Figs 1–2, 29–32.
Sample: 2017_KE-8-04.
Diagnostic: Apertures hexagonal; setae arising slightly within valve margin, base narrow, then usually widened and later tapered.
Comments: *Chaetoceros atlanticus* is an oceanic species [98]. Hasle and Syvertsen [19] regarded it as a cosmopolitan species; it has mainly been recorded in northern and southern cold-water regions (e.g., [98,100]. This species may be transported along the Oyashio Current.

***Chaetoceros didymus* Ehrenberg, 1845 [Figure 8c (LM)]**

References: Ishii et al. [101], Figs 22, 23; Hasle and Syvertsen [19], p. 207, pl. 43; Lee et al. [96], p. 621, Figs 22–23, 85–88.
Sample: 2017_KE-8-04.

Distribution: *Chaetoceros didymus* is distributed in warm to temperate regions (Hasle and Syvertsen [19]. This species may be transported along the Kuroshio Current.

Comments: Although we only present the morphology of resting spores of *C. didymus*, Ishii, Iwataki, Matsuoka and Imai [101] proposed identification criteria for the resting spores of *Chaetoceros* species: The combination of morphological characteristics such as the single ring of puncta, the primary and secondary valve face, and mantle structure can help to identify the *Chaetoceros* species. Based on the resting spore morphology, the primary valve face is vaulted and smooth; the secondary valve face is inflated in the center, with paired setae emerging from the valve apex.

### *Chaetoceros eibenii* Grunow, 1882 [Figure 8d (LM)]

References: Koch and Rivera [102], p. 65, Figs 13–22; Hasle and Syvertsen [19], p. 201, pl. 41; Lee et al. 2014a, p. 181, Figs 19-20, 57-60.

Sample: 2017_KE-8-01.

Diagnostic: Large robust setae; sibling setae diverged at 90° angle.

Distribution: *Chaetoceros eibenii* is distributed in temperate to warm-water regions [19]. This species may be transported along the Kuroshio Current.

### *Chaetoceros messanensis* Castracane, 1875 [Figure 8e (LM)]

References: Hasle and Syvertsen [19], p. 216, pl. 45; Lee et al. [96], p. 623, Figs 29–30, 101–104.

Sample: 2017_KE-8-01.

Diagnostic: Special intercalary setae thicker than the others, first fused, then forked.

Distribution *Chaetoceros messanensis* is an oceanic warm-water species that mainly occurs in tropical and subtropical regions (e.g., [19,98]. This species may be transported along the Kuroshio Current.

### *Chaetoceros peruvianus* Brightwell, 1856 [Figure 8f (LM)]

References: Cupp [98], p. 113, Fig 68a–c; Hasle and Syvertsen [19], p. 195, pl. 38; Lee et al. [99], p. 184, Figs 21–22, 61–64.

Sample: 2017_KE-8-01.

Diagnostic: Heterovalvate; setae of upper valve arise in pervalvar direction from near valve center, abut with groove between them, turn sharply, and run backward in more or less outwardly convex curves.

Distribution: *Chaetoceros peruvinavus* is distributed in warm to temperate regions [19]. This species may be transported along the Kuroshio Current.

### *Chaetoceros radicans* F. Schütt, 1895 [Figure 8g (LM)]

References: Cupp [98], p. 141, Fig 97; Hasle and Syvertsen [19], p. 213, pl. 45; Lee et al. [96], p. 625, Figs 36–38, 113–116.

Sample: 2017_KE-8-01.

Dimensions: Valve 14 μm in apical axis.

Diagnostic: Intercalary setae have hair-like siliceous spines.

Distribution: *Chaetoceros radicans* is a cosmopolitan species [19]. Due to its occurrence in a wide range of temperatures, it is difficult to determine which current is related to the occurrence of *C. radicans* from the KE region.

**Class Bacillariophyceae Haeckel, 1878**
**Subclass Fragilariophycidae Round in Round et al., 1990**
**Order Thalassionematales Round in Round et al., 1990**
**Family Thalassionemataceae Round in Round et al., 1990**
**Genus *Lioloma* Hasle, 1996**
***Lioloma pacificum* (Cupp) Hasle, 1996 [Figure 9a–c (LM)]**

Basionym: *Thalassiothrix mediterranea* var. *pacifica* Cupp 1943.

References: Cupp [98] p. 185, Fig 136; Simonsen [20], p. 38, pl. 24, Fig 5; Hasle and Syvertsen [19], p. 254.

Sample: 2017-KE-01.

Dimensions: 667.9 μm long, 6 μm wide; 13 areolae in 10 μm.

Diagnostic: Valve slightly expanded in the middle and almost the same size as the head pole until it becomes somewhat enlarged for a short distance about one-third from wedge-shaped, blunt-ended foot pole.

Distribution: *Lioloma pacificum* is distributed in temperate to warm-water regions [19]. This species may be transported along the Kuroshio Current.

**Genus *Thalassionema* Grunow ex Mereschkowsky, 1902**

***Thalassionema bacillare* (Heiden) Kolbe, 1955 [Figure 9d,e (LM)]**

Basionym: *Spinigera bacillaris* Heiden, 1928.

Synonym: *Thalassionema elegans* Hustedt, 1958.

References: Hasle and Syvertsen [19], p. 262.

Sample: 2017-KE-01.

Dimensions: 89.6–194.6 μm long, 2.9 μm wide.

Diagnostic: Valves more or less inflated in the middle and slightly expanded apices. Areolae visible with LM as circular or subcircular holes, a structure sometimes visible within the holes.

Distribution: *Thalassionema bacillare* is distributed in warm-water regions [19]. This species may be transported along the Kuroshio Current.

***Thalassionema frauenfeldii* (Grunow) Tempère & Peragallo, 1910 [Figure 9f–j (LM)]**

Basionym: *Asterionella frauenfeldii* Grunow, 1863.

Synonym: *Thalassiothrix frauenfeldii* (Grunow) Grunow, 1880.

References: Hasle and Syvertsen [19], p. 262.

Sample: 2017-KE-01, 2017-KE-02.

Dimensions: 111.7–350.2 μm long, 2.5–4.9 μm wide; six to ten areolae in 10 μm.

Diagnostic: Valves more or less expanded in the middle part, and slightly expanded apices. Areolae crossed by a simple strongly silicified bar, discernible with LM.

Distribution: *Thalassionema frauenfeldii* is distributed in temperate to warm-water regions [19]. This species may be transported along the Kuroshio Current.

***Thalassionema kuroshioensis* Sugie & K.Suzuki, 2015 [Figure 9o,p (LM)]**

References: Sugie and Suzuki [103], p. 240, Figs 2–21.

Sample: 2017-KE-02.

Dimensions: 487.3 μm long, 4.4 μm wide; four areolae in 10 μm.

Diagnostic: Valves linear with isopolar rounded apices.

Distribution: Sugie and Suzuki [103] found *T. kuroshioensis* from 28° to 33° N, 138° E in the north Pacific. This species may be transported along the Kuroshio Current.

***Thalassionema nitzschioides* var. *nitzschoides* (Grunow) Grunow, 1902 [Figure 9k–m (LM)]**

References: Hasle and Syvertsen [19], p. 262.

Sample: 2017-KE-02.

Dimensions: 41.1–59.7 μm long, 4.0–5.1 μm wide; eight to eleven areolae in 10 μm.

Diagnostic: Valves linear to narrowly lanceolate, with rounded to capitate apices; marginal structure visible with LM as ribs.

Distribution: *Thalassionema nitzschioides* is a cosmopolitan species but is not abundant in the polar regions [19]. This species may be transported along the Kuroshio Current.

***Thalassionema nitzschioides* var. *parva* Heiden & Kolbe, 1928 [Figure 9n (LM)]**

References: Lange-Bertalot [104], pl. 137, Fig 5; Siqueiros-Beltrones et al. [105], Fig 8f.

Sample: 2017-KE-01.

Dimensions: 14.5 μm long, 4.1 μm wide.

Diagnostic: Valves short, linear with rounded isopolar apices.

Distribution: *Thalassionema nitzscioides* var. *parva* has been reported in tropical regions: Revillagigedo Archipelago, Mexico [105]; near the shore of North America [104]. This species may be transported along the Kuroshio Current.

**Subclass Urneidophycidae Medlin, 2016**
**Order Rhaphoneidales Round in Round et al., 1990**
**Family Rhaphoneidaceae Forti, 1912**
**Genus *Neodelphineis* H. Takano, 1983**
***Neodelphineis indica* (F.J.R. Taylor) Tanimura, 1992 [Figure 12a,b (SEM)]**

References: Hasle and Syvertsen [70], p. 309, Figs 35–41; Hasle and Syvertsen [19], p. 249.
Sample: 2017_KE-8-01; 2017_KE-8-02; 2017_KE-8-04.
Dimensions: 37.4–48.2 μm long, 7.0–9.3 μm wide; 15–17 striae in 10 μm.
Diagnostic: Valves linear, inflated at the center, and rounded apices.
Distribution: *Neodelphineis indica* is distributed in warm to temperate regions [19]. This species may be transported along the Kuroshio Current.

**Subclass Bacillariophycidae D.G.Mann in Round et al., 1990**
**Order Naviculales Bessey, 1907**
**Family Naviculaceae Kützing, 1844**
**Genus *Navicula* Bory, 1822**
***Navicula* cf. *transitantoides* Foged, 1975 [Figure 10a–d (LM)]**

References: Foged [35], p. 44, pl. 19, Figs 10, 11; Witkowski et al. [106], p. 310, pl. 135, Figs 2–6, pl. 146, Fig 13; Al-Handal et al. [107], p. 128, Fig 53.
Sample: 2017_KE-8-01; 2017_KE-8-02; 2017_KE-8-04.
Dimensions: 37.4–48.2 μm long, 7.0–9.3 μm wide; 15–17 striae in 10 μm.
Diagnostic: Valves lanceolate with acute apices. Transapical striae parallel throughout the valve.
Distribution: *Navicula transitantoides* was first described from the east coast of Africa, Tanzania, in the Indian Ocean [35]. Subsequently, Witkowski, Lange-Bertalot and Metzeltin [106] observed it in European waters, and Al-Handal, Thomas and Pennesi [107] reported it in the Gulf, but the specimen from the Gulf had coarser striae than that from the original publication (11–12 vs. 14–15). Due to the lack of additional records along the boundary of the northwestern Pacific Ocean, it is difficult to determine which current is related to the occurrence of *N.* cf. *transitantoides* in the KE region.
Comments: The specimen from the northwestern Pacific Ocean is similar to *N. transitantoides* in the valve outline and striae arrangement, but the striae density is finer than in previous reports.

**Family Pleurosigmataceae Mereschowsky, 1903**
**Genus *Pleurosigma* W. Smith, 1852**
***Pleurosigma directum* Grunow, 1880 [Figure 10e,f (LM)]**

References: Simonsen [20], p. 45, pl. 29, Figs 2a,b; Hasle and Syvertsen [19], p. 282.
Sample: 2017_KE-8-06; 2017_KE-8-09.
Dimensions: 100.3–104.1 μm long, 16.8–18.0 μm wide; 18–19 transapical striae 10 μm in size; 20–23 longitudinal striae in 10 μm.
Diagnostic: Valves rhomboic–lanceolate; raphe almost straight.
Distribution: Hasle and Syvertsen [19] regarded *P. directum* as a probably cosmopolitan species. Due to the lack of additional records along the boundary of the northwestern Pacific Ocean, it is difficult to determine which current is related to the occurrence of *P. directum* in the KE region.

***Pleurosigma diversestriatum* F. Meister, 1934 [Figure 10g (LM)]**

References: Foged [35], p. 50, pl. 17, Fig 3; Foged [108], p. 119, pl. 22, Fig 6; Navarro 1982, p. 325, Figs 93–94; Cardinal et al. 1989, Fig 50; Sterrenburg 2001a, p. 124, Figs 6, 15–18.

Sample: 2017_KE-8-02.

Dimensions: 87.7 µm long, 17.6 µm wide; 17 transapical striae in 10 µm; 12 longitudinal striae in 10 µm.

Diagnostic: Valve sigmoid with acutely rounded apices; striae throughout in transverse and two oblique systems, the latter strongly curved in the central portion of the valve so that they are markedly stronger centrally.

Distribution: *Pleurosigma diversestriatum* was originally described in Nagasaki, Japan, in the northwestern Pacific Ocean (Meister 1934). This species was also reported in Tanzania, the Indian Ocean (Foged 1975), Australia (Foged 1978), and the Indonesian Archipelago (Sterrenburg 2001). This species may be transported along the Kuroshio Current.

**Order Bacillariales Hendey, 1937**
**Family Bacillariaceae Ehrenberg, 1831**
**Genus *Alveus* Kaczmarska & Fryxell, 1996**
***Alveus marinus* (Grunow) Kaczmarska & Fryxell, 1996 [Figure 10h,i (LM)]**

Basionym: *Nitzschia angustata* var. *marina* Grunow, 1878.

Synonyms: *Nitzschia marina* (Grunow) Grunow in Cleve & Grunow, 1880; *Synedra gausii* Heiden in Heiden & Kolbe, 1928.

References: Kaczmarska and Fryxell [109], p. 3, Figs 1–35.

Sample: 2017_KE-8-02.

Dimensions: 225.7 µm long, 12.6 µm wide; 10 transapical striae in 10 µm; 15 areolae in 10 µm; nine fibulae in 10 µm.

Diagnostic: Valve robust and linear with rounded apices; striae biseriate, parallel throughout.

Distribution: *Alveus marinus* is distributed in warm-water regions [19]. This species may be transported along the Kuroshio Current.

**Genus *Fragilariopsis* Hustedt, 1913**
***Fragilariopsis doliolus* (Wallich) Medlin & P.A. Sims, 1993 [Figure 10j,k (LM)]**

Basionym: *Synedra doliolus* Wallich 1860, p. 48, pl. 2, Fig 19.

Synonyms: *Pseudo-eunotia doliolus* (Wallich) Grunow in Van Heurck 1880, pl. 35, Fig 22.

References: Hasle and Syvertsen [19], p. 303, pl. 69.

Sample: 2017_KE-8-01; 2017_KE-8-06; 2017_KE-8-09; 2017_KE-8-18.

Dimensions: 47.5–80.3 µm long, 7.4–10.3 µm wide; 10–13 transapical striae in 10 µm; 16–22 areolae in 10 µm; 10–11 fibulae in 10 µm.

Distribution: *Fragilariopsis doliolus* is a warm-water species [19]. This species may be transported along the Kuroshio Current.

***Fragilariopsis* aff. *oceanica* (Cleve) Hasle, 1965 [Figure 10l (LM)]**

References: Hasle and Syvertsen [19], p. 299, pl. 67; Lundholm and Hasle [110], p. 442, Figs 1–23.

Sample: 2017_KE-8-02.

Dimensions: 39.6 µm long, 7.6 µm wide; 13 transapical striae in 10 µm.

Diagnostic: Valve narrowly elliptic and isopolar with lanceolate apices; striae parallel to middle and radiate slightly at the ends.

Comments: *Fragilariopsis oceanica* has mainly been reported in northern cold-water regions [19]. In their study on the diatom fluxes in the subarctic Pacific, Onodera et al. [111] did not find *F. oceanica* below 40N. Although the valve outline and dimensions of the KE specimens match *F. oceanica*, their limited occurrence in the Arctic region casts doubt on its certain identification as *F. oceanica*. We temporarily identified the species as *Fragilariopsis* aff. *oceanica*.

**Genus *Neodenticula* Akiba & Yanagisawa, 1986**
***Neodenticula seminae* (Simonsen & T. Kanaya) Akiba & Yanagisawa, 1986 [Figure 10m,n (LM)]**

Basionym: *Denticula seminae* Simonsen & T. Kanaya 1961, p. 503, pl. 1, Figs 26–30.

Synonyms: *Denticulopsis seminae* (Simonsen & T. Kanaya) Simonsen 1979, p. 65; *Denticula maina* Semina 1956, p. 82, Figs 1, 2.

References: Hasle and Syvertsen [19], p. 306, pl. 69; Poulin et al. [112], p. 129, 1–54.

Sample: 2017_KE-8-02.

Dimensions: 20.3–33.2 μm long, 5.9–7.8 μm wide; seven to eight transapical striae in 10 μm; seven to eight fibulae in 10 μm.

Diagnostic: Valves linear to elliptical with broadly rounded ends; striae straight; pseudosepta formed distinct lines; both ends of a crossbar connected to the thick valve wall by a suture.

Distribution: *Neodenticula seminae* is common in the North Pacific Ocean (Hasle & Syvertsen 1996) and has also been observed in the Gulf of St. Lawrence, Canada, in the North Atlantic [112]. This species may be transported along the Oyashio Current.

### Genus *Nitzschia* Hassall, 1845
### *Nitzschia bicapitata* Cleve, 1901 [Figure 10o–s (LM), Figure 12c,d (SEM)]

References: Fryxell [113], p. 46, Figs 1–11; Kaczmarska and Fryxell [114], p. 238, Figs 4–7, 9 and 10; Hasle and Syvertsen [19], p. 330, pl. 74.

Sample: 2017_KE-8-01; 2017_KE-8-02; 2017_KE-8-06; 2017_KE-8-09; 2017_KE-8-18.

Dimensions: 4.4–50.6 μm long, 2.5–6.84 μm wide; 24–40 transapical striae in 10 μm.

Diagnostic: Valves highly variable such as elliptical–lanceolate, linear–lanceolate, linear–elliptical with capitate apices. Striae arched; uniseriate throughout.

Distribution: *Nitzschia bicapiatata* is known as a warm to temperate water species [19]. This species may be transported along the Kuroshio Current.

### *Nitzschia bifurcata* Kaczmarska & Licea, 1986 [Figure 12e (SEM)]

References: Kaczmarska and Fryxell [114], p. 241, Figs 1–3, 8 A–E; Tanimura 1992, p. 138, Figs 9.8–9.10, 12.5–12.6.

Sample: 2017_KE-8-01.

Dimensions: 10.9 μm long, 4.5 μm wide; 36 transapical striae in 10 μm.

Diagnostic: Valves rhomboid with rostrate apices. Striae arched; uniseriate changes to biseriate near the margin opposite the raphe.

Distribution: *Nitzschia bifurcata* was first described in Gulf Stream materials (Kaczmarska and Fryxell 1986). Tanimura (1992) observed this species in sediment samples in the northwest Pacific, in the path of the Kuroshio Current.

### *Nitzschia interruptestriata* Simonsen, 1974 [Figure 10t,u (LM)]

References: Simonsen [20], p. 52, pl. 36, Figs 9–11, pl. 37, Figs 1–7, pl. 38, Figs 1–12; Barron [115], pl. 6, Figs 13, 19.

Sample: 2017_KE-8-04; 2017_KE-8-06.

Dimensions: 74.0–89.6 μm long, 4.7–6.6 μm wide; 16–27 transapical striae in 10 μm; 7–12 fibulae in 10 μm.

Diagnostic: Valves lanceolate with rostrate apices; striae parallel throughout, interrupted by the apical hyaline line near the middle part of the valve.

Distribution: Simonsen [20] found *N. interruptestirata* in the Indian Ocean, and Barron [115] reported this species from sediment samples in the tropical eastern Pacific. Tanimura [116] found this species in a sediment trap deployed in the path of the Kuroshio Current of the northwest Pacific. This species may be transported along the Kuroshio Current.

### *Nitzschia kolaczeckii* Grunow, 1867 [Figure 10v,w (LM)]

References: Schmidt [117], pl. 349, Figs 38, 39; Foged [35], p. 46, pl. 28, Figs 7, 8; Koizumi [118], Fig 25; Hasle and Syvertsen [19], p. 328.

Sample: 2017_KE-8-01; 2017_KE-8-02.

Dimensions: 87.6–96.9 μm long, 8.1–8.2 μm wide; 14–16 transapical striae in 10 μm; 10–12 areolae in 10 μm; seven to eight fibulae in 10 μm.

Diagnostic: Valve lanceolate with slightly stretched apices; areolae distinct with diagonal grid pattern.

Distribution: *Nitzschia kolaczeckii* is distributed in warm-water regions [19]: the Philippines [117]; East Sea (Sea of Japan) [118]. This species may be transported along the Kuroshio Current.

### *Nitzschia sicula* **var.** *sicula* **(Castracane) Hustedt, 1958 [Figure 10x (LM)]**

Basionym: *Synedra sicula* Castracane, 1875.
References: Hasle and Syvertsen [19], p. 327; Tanimura [116], Fig 10.1.
Sample: 2017_KE-8-01.
Dimensions: 33.2 μm long, 7.7 μm wide; seven transapical striae in 10 μm.
Diagnostic: Valve lanceolate with obtusely rounded apices. Transapical striae parallel throughout.
Distribution: *Nitzschia sicula* var. *sicula* is distributed in warm-water to temperate regions [19]. This species may be transported along the Kuroshio Current.

### *Nitzschia sicula* **var.** *bicuneata* **(Grunow) Hasle, 1960 [Figure 10y–aa (LM)]**

Basionym: *Rhaphoneis bicuneata* Grunow in Cleve and Möller, 1879.
References: Hasle and Syvertsen [19], p. 327, pl. 75.
Sample: 2017_KE-8-01; 2017_KE-8-05; 2017_KE-8-09.
Dimensions: 23.8–44.0 μm long, 6.1–9.8 μm wide; 7–10 transapical striae in 10 μm; 7–10 fibulae in 10 μm.
Diagnostic: Valve lanceolate with slightly rounded apices; transapical striae in the central area are rather short due to the presence of a central depression.
Distribution: *Nitzschia sicula* var. *bicuneata* is distributed in warm-water to temperate regions [19]. This species may be transported along the Kuroshio Current.

### **Genus** *Psammodictyon* **D.G.Mann in Round et al., 1990**
### *Psammodictyon* **sp. [Figure 10ab,ac (LM), 140 (SEM)]**

Sample: 2017_KE-8-01; 2017_KE-8-02.
Dimensions: 14.9–16.5 μm long, 6.7–6.9 μm wide; transapical striae invisible in LM; 13–16 fibulae in 10 μm.

### **Genus** *Pseudo-nitzschia* **H. Peragallo, 1900**
### *Pseudo-nitzschia turgiduloides* **(Hustedt) Hasle, 1993 [Figure 10ad (LM)]**

Basionym: *Nitzschia turgiduloides* Hasle, 1996.
Synoynm: *Pseudo-nitzschia barkleyi* var. *obtusa* Manguin, 1960.
References: Hasle and Syvertsen [19], p. 319; Almandoz et al. [119], p. 440, Figs 7a–d.
Sample: 2017_KE-8-01; 2017_KE-8-02.
Dimensions: 129.9 μm long, 2.9 μm wide; 20 transapical striae in 10 μm; 11 fibulae in 10 μm.
Diagnostic: Valve linear, expanded in the middle with rounded apices; interstriae visible in LM; central space distinct.
Distribution: *Pseudo-nitzschia turgiduloides* is a commonly known southern cold-water species [19,119]. The occurrence of this species was first identified in the northwest Pacific Ocean. This species may be transported along the Oyashio Current.

### **Genus** *Tryblionella* **W. Smith, 1853**
### *Tryblionella coarctata* **(Grunow) D.G. Mann, 1990 [Figure 10ae,af (LM)]**

Basionym: *Nitzschia coarctata* Grunow, 1880.
References: Cleve and Grunow [120], p. 68; Witkowski, Lange-Bertalot and Metzeltin [106], p. 374, pl. 183, Fig 13, pl. 186, Figs 4–13; Al-Handal et al. [121], p. 36, pl. 10, Fig 13; López-Fuerte, Siqueiros Beltrones and Altamirano-Cerecedo [48], p. 20, Figs 43a–d,g,h.
Sample: 2017-KE-02.
Dimensions: 26.2–34.0 μm long, 10.2–12.5 μm wide; 15–16 transapical striae in 10 μm; 12 fibulae in 10 μm.

Diagnostic: Valve panduriform with broadly cuneate, short rostrate apices; transapical striae distinct, interrupted by longitudinal fold.

Distribution: *Tryblionella coarctata* is a neritic species, known as a marine to brackish-water species [106], and it has mainly been reported in temperate to tropical regions: off Santa Rosalia, Baja California Sur, Mexico [48]. This species may have been transported from the coastal areas to the KE region via the Kuroshio Current.

Comments: The KE specimen exhibits denser striae (15–16 in 10 μm) compared to the original and subsequent works: Cleve and Grunow [120] originally described a striae density of 11–12 in 10 μm; Witkowski, Lange-Bertalot and Metzeltin [106] noted a sparser arrangement than the original material (8–10 in 10 μm); López-Fuerte, Siqueiros Beltrones and Altamirano-Cerecedo [48] described a striae density similar to the original material (12 in 10 μm). The KE specimen shows similarity to the specimen from Reunion and Rodrigues Islands as described by Al-Handal, Compere and Riaux-Gobin [121], who also reported a striae density of 12–16 in 10 μm.

## 4. Discussion

The development of taxonomic reference flora of diatoms from specific regions is crucial for bolstering long-term records [122], facilitating the accurate identification of species [123], and evaluating diatom assemblage structure efficiently [124,125]. This is the first study that evaluates the flora of diatoms in the KE region; we identified and described 82 diatom species. The most diverse genera were *Thalassiosira* (10 spp.) and *Chaetoceros* (7 spp.), which were commonly reported in pelagic systems, but also found in coastal regions (e.g., [84,96,99]. The occurrences of these genera in both neritic and pelagic systems indicate that they can be distributed worldwide via the current, but the species in the KE are mainly composed of taxa that prefer pelagic habitats. Based on the previous species records provided in the taxa list in the Results section, we can divide the diatoms into cold, warm, and eurythermal species. Fifty-five species were warm-water taxa that were transported via the Kuroshio Current, while twelve cold-water species were transported via the Oyashio Current (Table 1). The transfer of the remaining 14 species was uncertain in terms of which current influenced them, because they were widely distributed from cold- to warm-water regions. In the KE region, warm-water species predominated along the path of the Kuroshio Current, but the occurrence of species migrating southward along the Oyashio Current indicates that this region can be the crossroad between warm- and cold-water species. Along the Kuroshio Current, several studies on diatoms were performed near the east of Japan (e.g., [116,126,127]). Tanimura [116] reported 124 species except for the unidentified taxa from the time-series sediment trap in the Japanese Trench, and he illustrated results for only 37 taxa. Koizumi, Irino and Oba [126] and Koizumi and Yamamoto [127] used the diatoms as a proxy of a paleography of the Kuroshio Current, and they listed 145 diatom species, except for unidentified ones, and did not provide any flora. From these three studies, 209 diatom species were listed from the northern part of the Kuroshio Current (NKC) (Table S1). These works showed a higher number of species than our work, due to their study areas being close to Japanese coasts and including a greater number of neritic species than the KE regions. Of 81 taxa, excluding *Psammodictyon* sp., which occurred in the present study, 51 are also reported in the compiled checklist of the NKC, but 30 species, namely *Coscinodiscopsis jonesaina*, *Coscinodiscus argus*, *C. gigas*, *Actinocyclus iraidae*, *Azpeitia neocrenulata*, *Proboscia indica*, *Sundstroemia pungens*, *Detonula confervacea*, *Minidiscus trioculatus*, *Planktoniella blanda*, *Shionodiscus* cf. *oestrupii* var. *venrickae*, *S.* aff. *poro-irregulatus*, *S. variantus*, *Takanoa bingesis*, *Thalassiosira mendiolana*, *T. punctifera*, *T. sacketii*, *Lithodesmium variabile*, *Cerataulina pelagica*, *Bacterastrum furcatum*, *Chaetoceros atlanticus*, *C. eibenii*, *Lioloma pacificum*, *Thalassionema kuroshioensis*, *Navicula* cf. *transistantoides*, *Pleurosigma directum*, *P. diverse-striatum*, *Pseudo-nitzschia turgiduloides*, *Fragilaria* aff. *oceanica*, and *Tryblionella coarctata*, were not listed. With the addition of our flora, a preliminary checklist reports 239 diatom taxa occurring in the northeastern path of the Kuroshio Current. The diatom flora and future continued research in the KE area

will help us understand the hydrographical dynamics of the Kuroshio Current through the emergence of additional coastal species that can migrate along the current.

**Supplementary Materials:** The following supporting information can be downloaded at: https://www.mdpi.com/article/10.3390/taxonomy4030025/s1, Table S1: The preliminary checklist of the diatoms in the pathway of Kuroshio Current. The species in bold indicate newly reported in this study.

**Author Contributions:** Conceptualization, J.S.P.; methodology, J.S.P. and H.J.H.; validation, J.S.P. and H.J.K.; formal analysis, J.S.P.; investigation, J.S.P., H.J.K., K.-W.L. and Y.J.K.; resources, H.J.K.; data curation, J.S.P.; writing—original draft preparation, J.S.P.; writing—review and editing, J.S.P., H.J.K., K.-W.L. and Y.J.K.; visualization, J.S.P. and H.J.H.; supervision, J.S.P.; funding acquisition, Y.J.K. All authors have read and agreed to the published version of the manuscript.

**Funding:** This research was supported by the High Seas bioresources program of the Korea Institute of Marine Science & Technology Promotion (KIMST) funded by the Ministry of Oceans and Fisheries (KIMST-20210646).

**Data Availability Statement:** All data generated or analyzed during this study are included in this published article.

**Conflicts of Interest:** The authors declare no conflict of interest.

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
