# Peer review of "Diatom Flora from Time-Series Sediment Trap in the Kuroshio Extension Region of the Northwestern Pacific"

_2673-6500, doi:10.3390/taxonomy4030025_

Round 1
Reviewer 1 Report
Comments and Suggestions for Authors
PLEASE SEE ATTACHED FILE

Comments on the Quality of English LanguageALTHOUGH THE WRITING IS SOMEWHAT CLEAR THAT ARE SYNTAX AND SEMANTIC ISSUES THAT REQUIRE CORRECTION. SHOULD BE PROPERLY ATTENDED
Author Response
Reviewer:1
Title
Line 1. SUGGEST TITLE CHANGE. FIRST AVOID USING THE TERM VOUCHER. IT IS IMPRECISE. SUBSTITUTE BY OTHERS BEING MORE COMPROMISING.
We eliminated the word ‘voucher’ as mentioned reviewer 1.
Abstract
Line 11-14. THIS PARAGRAPH IS CONFUSING AND NEEDS TO BE REWRITEN.
We revised the paragraph as “Precise identification of the diatom species is fundamental to correctly interpreting their roles in the marine ecosystems, the documentation of the species records with illustration, therefore is essential to guarantee the ecological works and the continuous use of the compositional data in future works.”
Line 20. CHANGE PHRASE TO: transported by the Oyashio and Kuroshio currents, respectively
We changed the phrase as mentioned reviewer 1.
Introduction
LINES 45-50. THERE SEEMS TO BE NO NEED TO ADDRESS THIS ISSUE, INASMUCH IT DOES NOT SUPPORT THE AIM OF THE STUDY. BESIDES, AS A MORPHOLOGIST I AM FAR FROM BEING DISCOURAGED. I ACTUALLY THINK AS IN THE FINAL PART OF THE PARAGRAPH
We eliminated the paragraph as mentioned reviewer 1.
Line 51. THE USE OF DATA DERIVED FROM SEDIMENT TRAPS NEEDS MORE SUPPORT, e.g., Rochín Bañaga, H., D. A. Siqueiros Beltrones & Jörg Bollmann. 2015. Benthic diatoms from shallow environments deposited at a 300 m depth in a southern Gulf of California basin. CICIMAR-Oceánides, 30(1): 71-76. doi.org/10.37543/ oceanides.v30i1.145
We rewrote the paragraph and cited the reference.
Line 53. AGAIN, THE TERM VOUCHER FLORA MAKES NO SENSE. BETTER TO REFER TO FLORISTICS AND ITS LINK TO ECOLOGY AND BIOGEOGRAPHY.
We eliminate the word “voucher” in the manuscript.
Line 57. AFTER REMOVING THE ABOVE INFORMATION, AND INCLUDING MORE BACKGROUND, WORK MORE ON YOUR PROBLEM THAT HAS TO DO WITH THE KUROSHIO EXTENSION! AS THE TITLE INDICATES! TRY RESCUING YOUR HYPOTHESIS, I.E., WHAT WERE YOU EXPECTING TO FIND AND ON WHAT GROUNDS?
We re-arranged the paragraph in introduction and clearly rewrote the purpose of the study as mentioned by reviewer.
Materials and methods
Line 70. FIGURE 1 NEEDS A LEYEND
We added legends for all figures.
Results
Line 89. AFTER CONSTRUCTING YOUR HYPOTHESIS, INDICATE IF IT IS BACKED OR IS IT REFUTED. ALSO, MENTION HERE YOUR TABLE (WITH NUMBER) DESCRIBING ALL THE DATA OR INFORMATION IT CONDENSES (AS IN THE ABSTRACT BUT EXTENDED). AND MOVE IT HERE, BEFORE THE PLATES.
The table was moved the front of the plate and cited in the result section.
Line 90. THESE PLATES SHOULD BE MOVED TO AFTER THE LIST WITH DESCRIPTIONS. MAYBE AS AN APPENDIX, AFTER LITERATURE CITED.
We rearranged the plates according to the species description.
Line 108. THIS TABLE CONSTITUTES IMPORTANT PART OF YOUR RESULTS... IT NEEDS A HEADING. AND NUMBERING, AND TO BE REFERED IN THE TEXT (RESULTS AND DISCUSSION) ALSO, IT SHOULD BE INSERTED AFTER TABLE 1 AND BEFORE THE PLATES.
The table was moved the front of the plate and cited in the result section.
Line 1157. THIS TABLE SHOULD BE MOVED RIGHT AFTER THE FIRST PARAGRAPH OF RESULTS.
The table was moved the front of the plate and cited in the result section.

Reviewer 2 Report
Comments and Suggestions for Authors
The manuscript entitled Diatom voucher flora from time-series sediment trap in the Kuroshio Extension region of the northwestern Pacific by Park et al recorded diatom species found in a sediment trap deployed in the Kuroshio Extension region, northwestern Pacific. The study provides valuable diatom data in the study region, providing a baseline of understanding of environment and ecosystem changes in the Kuroshio Extension region. In general, the manuscript is well written. Considering the valuable data and images, I would suggest moderate revision before acceptance of the manuscript.
A few specific comments:
1) Line 12, ‘,’ between ‘… correctly interpreting the marine ecosystems’ and ‘the documentation of the species records’ should be ‘.’
2) Line 124, is Yokoham a typo of Yokohama?
3) Line 243, ‘Arctic Sea’ do you mean ‘Arctic Ocean’?
4) Line 395, East ‘China’ Sea
5) Line 493, should be S. oestrupii var. venrickae
6) Line 611, is Thalassiosira mendiolana related to Fig. 131 and 132? Otherwise, which species is Fig. 131 related?
7) Line 707-714, I do not understand if the authors are talking about Hemiaulus sinensis or H. hauckii?
8) Fig. 74 is duplicated, one of which should be Fig. 73.
9) Fig. 76-85 are missing, while Fig. 69-75 are duplicated
10) For Coscinodiscus asteromphalus, the image in Fig. 5 is not a typical one without prominent larger central rosette.
11) For Coscinodiscus marginatus, the images in Fig. 7 and 8 are doubtful. Normally C. marginatus is much larger (35-145 μm in diameter; see Sancetta, Micropaleontology, 1987, 33(3), 230-241), while it is very small in this study (21.5-21.7 μm in diameter).
12) For Fragilariopsis cf. oceanica, it is hard to believe that this sea ice/cold water related specie is found at 33° N. In Onodera et al. (Deep-Sea Research II, 2005; BTW, this is a relevant work that the authors should read and cite), F. oceanica is recorded in stations KNOT and 50N, but absent in station 40N. This species should be something else.
Author Response
Reviewer: 2
Comments and Suggestions for Authors
The manuscript entitled Diatom voucher flora from time-series sediment trap in the Kuroshio Extension region of the northwestern Pacific by Park et al recorded diatom species found in a sediment trap deployed in the Kuroshio Extension region, northwestern Pacific. The study provides valuable diatom data in the study region, providing a baseline of understanding of environment and ecosystem changes in the Kuroshio Extension region. In general, the manuscript is well written. Considering the valuable data and images, I would suggest moderate revision before acceptance of the manuscript.
A few specific comments:
1) Line 12, ‘,’ between ‘… correctly interpreting the marine ecosystems’ and ‘the documentation of the species records’ should be ‘.’
We rewrote the paragraph as “Precise identification of the diatom species is fundamental to correctly interpreting their roles in the marine ecosystems, the documentation of the species records with illustration, therefore is essential to guarantee the ecological works and the continuous use of the compositional data in future works.”
2) Line 124, is Yokoham a typo of Yokohama?
Thank you for comment. A typo has been corrected.
3) Line 243, ‘Arctic Sea’ do you mean ‘Arctic Ocean’?
Thank you for comment. A typo has been corrected.
4) Line 395, East ‘China’ Sea
Thank you for comment. A typo has been corrected.
5) Line 493, should be S. oestrupii var. venrickae
Thank you for comment. A typo has been corrected.
6) Line 611, is Thalassiosira mendiolana related to Fig. 131 and 132? Otherwise, which species is Fig. 131 related?
The figure 131 is Thalassiosira diporocyclus.
7) Line 707-714, I do not understand if the authors are talking about Hemiaulus sinensis or H. hauckii?
Thank you for comment. The species is Hemiaulus sinesis based on the distinct areolae and rimoportula. A typo has been corrected.
8) Fig. 74 is duplicated, one of which should be Fig. 73.
Thank you for comment. The figure number has been corrected.
9) Fig. 76-85 are missing, while Fig. 69-75 are duplicated
Thank you for comment. The figures has been corrected.
10) For Coscinodiscus asteromphalus, the image in Fig. 5 is not a typical one without prominent larger central rosette.
Thank you for comment. As pointed out the reviewer, the species re-identified as C. argus. The obscure central rosette and the presence of valve face rimoportula match to C. argus rather than C. asteromphalus. We rewrote species information in line 152.
11) For Coscinodiscus marginatus, the images in Fig. 7 and 8 are doubtful. Normally C. marginatus is much larger (35-145 μm in diameter; see Sancetta, Micropaleontology, 1987, 33(3), 230-241), while it is very small in this study (21.5-21.7 μm in diameter).
Thank you for comment. As mentioned by reviewer, the species is temporally regarded Coscinodiscus cf. marginatus. And the reason is written in the comment in Line 203.
12) For Fragilariopsis cf. oceanica, it is hard to believe that this sea ice/cold water related specie is found at 33° N. In Onodera et al. (Deep-Sea Research II, 2005; BTW, this is a relevant work that the authors should read and cite), F. oceanica is recorded in stations KNOT and 50N, but absent in station 40N. This species should be something else.
We absolutely agree to reviewer’s comment. Although the valve outline and dimension match to F. oceanica, We hesitated the certain identification as F. oceanica due to the ecological distribution as mentioned reviewer. We regarded the species as Fragilariopsis aff. oceanica and wrote the reason in comment in Line 1012.

Round 2
Reviewer 2 Report
Comments and Suggestions for Authors
The manuscript has been improved according the advices and comments by the reviewers. I would prefer to see a section of Conclusion at the end of the manuscript. I am not sure if it is necessary for papers in such type. Otherwise, I would suggest to accept it as is.
Comments on the Quality of English LanguageMinor editing of English language.